# LANGUAGE MODELING VIA STOCHASTIC PROCESSES

**Rose E. Wang, Esin Durmus, Noah Goodman, Tatsunori B. Hashimoto**
Stanford University
{rewang, edurmus, ngoodman,thashim}@stanford.edu

## ABSTRACT

Modern language models can generate high-quality short texts. However, they often meander or are incoherent when generating longer texts. These issues arise from the next-token-only language modeling objective. To address these issues, we introduce Time Control (TC), a language model that implicitly plans via a latent stochastic process. TC does this by learning a representation which maps the dynamics of how text changes in a document to the dynamics of a stochastic process of interest. Using this representation, the language model can generate text by first implicitly generating a document plan via a stochastic process, and then generating text that is consistent with this latent plan. Compared to domain-specific methods and fine-tuning GPT2 across a variety of text domains, TC improves performance on text infilling and discourse coherence. On long text generation settings, TC preserves the text structure both in terms of ordering (up to $+40\%$ better) and text length consistency (up to $+17\%$ better). Human evaluators also prefer TC's output $28.6\%$ more than the baselines.[1]

## 1 INTRODUCTION

Large language models (LLM) such as GPT-2 have been extremely successful in text generation (Radford et al., 2019; Brown et al., 2020). However, LLMs are known to generate incoherent *long* texts. One reason is that they are unable to plan ahead or represent long-range text dynamics (Kiddon et al., 2016; Fan et al., 2019; Hua & Wang, 2020; Duboue & McKeown, 2001; Stent et al., 2004; Tamkin et al., 2020). As a result, they oftentimes produce wandering content with poor discourse structure and low relevance (Hua & Wang, 2020; Zhao et al., 2017; Xu et al., 2020); the text reads as if the model has no anchored goal when generating. These problems with coherence are further exacerbated when forcing autoregressive models to generate *longer* texts as the model struggles to extrapolate beyond its expected text end point. These problems suggest that LLMs currently fail to properly capture how documents evolve from beginning to end. Doing so is critical for succeeding in goal-oriented tasks such as story, dialog or recipe generation.

Prior work has explored the use of planning-based methods for generating globally coherent text (Kiddon et al., 2016; Fan et al., 2019; Hua & Wang, 2020; Duboue & McKeown, 2001; Stent et al., 2004). However, these methods rely on *manually* defining text dynamics for specific domains. Other work has attempted to use sentence representations for modeling text, such as with variational auto-encoders (Bowman et al., 2016) or contrastive learning (Gao et al., 2021; Devlin et al., 2019). Their shortcoming in text generation settings is that the latent representations are *static*: they capture semantic similarity between sentence neighbors, but don't capture how sentence embeddings evolve over a document. Methods including van den Oord et al. (2019) have tried to remedy this by learning a model of local latent dynamics. However, it is difficult to use learned local dynamics for generating accurate goal-conditioned trajectories, especially long-horizon ones. We explore an alternative that explicitly assumes a simple, fixed dynamics model with goal-conditioned generation.

In this work, we propose Time Control as a way to learn a latent space with known, goal-conditioned dynamics. We begin by assuming that meandering text generated without a goal can be represented as Brownian motion in latent space; this motion enforces the embeddings of neighboring sentences to be similar to each other, whereas those of distant sentences to be dissimilar. Goal-directed behavior can be incorporated into this model by conditioning on a fixed start and end point. In this

---

[1]The accompanying code can be found here: `https://github.com/rosewang2008/language_modeling_via_stochastic_processes`.

case, the Brownian motion becomes a Brownian bridge and the resulting latent trajectories abide by simple, closed-form dynamics.

In Time Control, we derive a novel contrastive objective for learning a latent space with Brownian bridge dynamics. We can then use this latent space to generate text that retains local coherence and has improved global coherence. To perform text generation, Time Control first plans a latent trajectory via the Brownian bridge process pinned at a start and end point. It then conditionally generates sentences using this latent plan. In our work, we decode latent plans by fine-tuning GPT2 to generate text conditioned on Time Control's latent trajectory. Trajectories from Time Control act as abstract semantic positions in a document that guide generation of fine-tuned language models.

In summary, our work's contributions are the following:

- We derive Time Control, a language model which explicitly models latent structure with Brownian bridge dynamics learned using a novel contrastive objective.

- Across a range of text domains, we show that Time Control generates more or equally coherent text on tasks including text infilling and forced long text generation, compared to task-specific methods.

- We validate that our latent representations capture text dynamics competitively by evaluating discourse coherence with human experiments.

- We ablate our method to understand the importance of the contrastive objective, enforcing Brownian bridge dynamics, and explicitly modeling latent dynamics.

## 2 RELATED WORKS

Generating long, coherent text is conceptually difficult for autoregressive models because they lack the ability to model text structure and dynamics (Lin et al., 2021). This means that they struggle to plan and look ahead which leads to generating globally incoherent text. Forcing autoregressive models to generate *longer* texts exacerbates this incoherence because the models struggle to extrapolate beyond their expected text end point. Prior work has tried to address the problem of generating globally coherent text with planning-based approaches (Puduppully et al., 2019; Moryossef et al., 2019; Fan et al., 2019; Kiddon et al., 2016). However, planning-based approaches rely on domain-specific heuristics for capturing text structure and dynamics.

Our work uses a contrastive objective to learn latent dynamics in text without domain-specific heuristics. Contrastive objectives have been applied to several domains, including language (Devlin et al., 2019; Iter et al., 2020; Liu & Liu, 2021), vision (Chen et al., 2020), and general time series data (Hyvarinen & Morioka, 2016; Hyvarinen et al., 2019). In particular for language, contrastive objectives have been applied to the next-sentence prediction task for improving BERT embeddings (Devlin et al., 2019) and to the discourse coherence setting (Nie et al., 2019; Chen et al., 2019b) for evaluating how coherent pairs of sentences are. However, these methods have two shortcomings which we address with our work. One is that the resulting sentence embeddings are often static: they capture semantic similarity between sentence neighbors, but don't capture how sentence embeddings evolve over a document. Two is that they are not used for generation and are limited to classification tasks like discourse coherence. Prior work has also tried fitting latent variable models (Bowman et al., 2016), however these generally result in poor language generation (He et al., 2018) or are domain-specific (Weber et al., 2020; Arora et al., 2016).

Our work is closely related to Contrastive Predictive Coding (CPC) from van den Oord et al. (2019). The key difference is CPC *implicitly* learns *unconditioned* latent dynamics, whereas we impose known goal-conditioned dynamics on our latent space. Doing so, we can extrapolate successfully further in time. Additionally, our method builds off of recent findings that contrastive objectives can be used to approximate local transition kernels of stochastic processes (Liu et al., 2021). The main difference between Liu et al. (2021) and our work is that they focus on provable conditions for latent recovery; we focus on empirically effective methods that leverage similar insights for recovering latent representations from language. Finally, our use of stochastic processes draws similarities to diffusion models (Song et al., 2020; Sohl-Dickstein et al., 2015) which apply a chain of diffusion steps onto the data and learn to reverse the diffusion process. However, our application

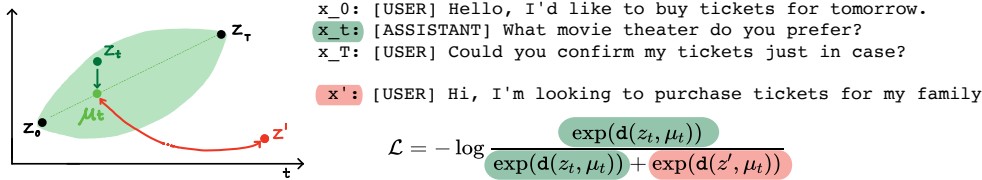

Figure 1: Latent space for a positive triplet of sentences $(x_0, x_t, x_T)$ that are part of the same conversation. Time Control maps positive triplets to a smooth Brownian bridge trajectory. It embeds $z_t$ close to the expected embedding $\mu_t$ pinned by $z_0, z_T$. The green oval area illustrates the uncertainty over $z_t$ as a function of how close $t$ is to 0 and $T$. In contrast, a negative random sentence $x'$ from a different conversation is not coherent with $x_0$ and $x_T$; thus, it is embedded far from $\mu_t$. This is captured by our contrastive loss, $\mathcal{L}$.

is conceptually different: diffusion processes characterize properties of our latent space and are not a fixed inference method in our work.

## 3 TIME CONTROL

The intuition behind Time Control is to learn a latent space with smooth temporal dynamics for modeling and generating coherent text. We detail Time Control in three sections. The first section discusses training the encoder via contrastive learning to map sentences to a Brownian bridge (Revuz & Yor, 2013) latent space. The second section discusses training a decoder to reconstruct sentences from this latent space. The third section discusses generating text from Time Control.

### 3.1 TRAINING AN ENCODER WITH BROWNIAN BRIDGE DYNAMICS

Our encoder is a nonlinear mapping from raw input space to latent space, $f_\theta : \mathcal{X} \to \mathcal{Z}$. The objective for the encoder is to map high-dimensional sequential data into low-dimensional latents which follow a stochastic process of interest—in this paper, it is the Brownian bridge process. The density of a Brownian bridge process between an arbitrary start point $z_0$ at $t = 0$ and end point $z_T$ at $t = T$ is,

$$p(z_t|z_0, z_T) = \mathcal{N}\left(\left(1 - \frac{t}{T}\right)z_0 + \frac{t}{T}z_T, \frac{t(T-t)}{T}\right). \tag{1}$$

This density is intuitive to understand: It acts like a noisy linear interpolation between the start and end point of the trajectory, where $z_t$ should be more like $z_0$ at the start and more like $z_T$ at the end of the trajectory. Uncertainty is highest in the middle region, and low near the end points (rf. Figure 1).

Consider a set of triplet observations, $(x_1, x_2, x_3)$. The goal of our work is to ensure that $f_\theta(x_1), f_\theta(x_2), f_\theta(x_3)$ follow the Brownian bridge transition density in Equation 1. We ensure this using a contrastive objective. Formally, given multiple sequences of data points, $X = \{x_1, ..., x_N\}$, we draw batches consisting of randomly sampled positive triplets $x_0, x_t, x_T$ where $0 < t < T$: $\mathcal{B} = \{(x_0, x_t, x_T)\}$.[2] Our encoder is optimized by,

$$\mathcal{L}_N = \mathbb{E}_X\left[-\log \frac{\exp(\mathrm{d}(x_0, x_t, x_T; f_\theta))}{\sum\limits_{(x_0, x_{t'}, x_T) \in \mathcal{B}} \exp(\mathrm{d}(x_0, x_{t'}, x_T; f_\theta))}\right], \text{where} \tag{2}$$

$$\mathrm{d}(x_0, x_t, x_T; f_\theta) = -\frac{1}{2\sigma^2}\bigg\|\underbrace{f_\theta(x_t)}_{z_t} - \underbrace{\left(1 - \frac{t}{T}\right)f_\theta(x_0) - \frac{t}{T}f_\theta(x_T)}_{\text{mean in Equation 1}}\bigg\|_2^2 \tag{3}$$

---

[2]We use indices $0, t, T$ to denote the start, middle and end point of a Brownian bridge, but these do *not* correspond to strictly sampling the first, middle and last sentence of a document. $x_0, x_t, x_T$ can be *any* sentence in a document as long as $x_0$ comes before $x_t$ and $x_t$ before $x_T$ in the document.

$\sigma^2$ is the variance in Equation 1: $\frac{t(T-t)}{T}$. Note that Equation 2 sums over negative middle contrasts, $x_{t'}$. This objective can be viewed as maximizing the extent to which true triplets from the data follow the Brownian bridge process while minimizing the extent to which an alternative mid-point sampled from another sequence does so. [3]

Figure 1 illustrates how the objective translates into the language setting for training the encoder. The objective samples triplet sentences from a document. Sentences drawn from the same document make up a smooth latent trajectory; they should be close to each other and follow the conditional density in latent space. Sentences drawn from different documents should not make up a smooth trajectory and should less likely follow bridge dynamics.

**Connection to mutual information estimation and triplet classification** We draw connections between our contrastive loss and the mutual information estimation setup from van den Oord et al. (2019); Poole et al. (2019) (as $|\mathcal{B}| \to \infty$) and the classification setup from Liu et al. (2021) ($|\mathcal{B}| = 1$).

Following van den Oord et al. (2019); Poole et al. (2019), this objective can be seen as a lower bound on the mutual information between the two end points and the middle point: $I(X_t, \{X_0, X_T\}) \geq \log(N) - \mathcal{L}_N$. Hence, by minimizing the contrastive loss, we are maximizing the amount of information between the trajectory and the linear interpolation of its end points.

Assuming $|\mathcal{B}| = 1$, we can draw a connection to the classification setup studied in Liu et al. (2021). They train a classifier to distinguish in- vs. out-of-order input pairs and show that the Bayes optimal logits for pair-wise classification can be written as a function of the stochastic process transition kernel. This is equivalent to the our loss on a single triplet $i$: $l_i = -\log \frac{\exp(\mathrm{d}(x_0, x_t, x_T; f_\theta))}{\exp(\mathrm{d}(x_0, x_t, x_T; f_\theta)) + \exp(\mathrm{d}(x_0, x_{t'}, x_T; f_\theta))}$. Liu et al. (2021) consider pairs whereas our work considers triplets; we show in Appendix A the pairwise and triplet setups are equivalent.

### 3.2 Training a decoder with latent plans

Here we discuss how to train a language model to decode latent plans for generation. We first map all the sentences in the training dataset to our learned latent space using the pretrained encoder $f_\theta$. This gives us a Brownian bridge trajectory of sentence-level latent codes $(z_0, \ldots, z_t, \ldots, z_T)$ for a document in the dataset. Then, rather than learning a decoder from scratch, we fine-tune GPT2 (Radford et al., 2019) to generate text conditioned on past context and the latent plan.

We fine-tune in the following manner. Let $x_1 \ldots x_W$ be a document with $W$ tokens and $T$ sentences used to train the decoder. Using the encoder $f_\theta$, we can obtain embeddings $z_1 \ldots z_T$ for each sentence. The decoder is a standard auto-regressive language model that is modified in the following way: at time $t$, the decoder must predict $x_t$ using all tokens in the past $x_{<t}$, as well as the sentence embedding $z_{s_t}$, where the index $s_t \in [T]$ is a map which takes each token to its corresponding sentence. This is a form of a reconstruction objective, as the identity of $x_t$ is encoded in $z_{s_t}$.

### 3.3 Generating text with latent plans at inference time

Figure 2 illustrates how the trained decoder generates text at inference time. Given two end points $z_0, z_T$, we sample a trajectory from a latent Brownian bridge, and then generate from the decoder conditioned on this bridge. In many situations, we may not know the endpoints of the Brownian bridge explicitly. In this case, we encode a set of sentences corresponding to start and end points (eg. the first and last sentences of our training set), and fit a Gaussian to these points to form a density estimate. Generating in this case involves first sampling from the Gaussian, and then generating as before from the bridge. More details on training and generation can be found in Appendix B.

## 4 Experiments

We now evaluate the ability of Time Control to capture text dynamics. Specifically, we aim to answer the following research questions (RQ):

---

[3]Empirically, we found Brownian bridge dynamics easier to recover with triplets rather than pairs of contrasts. Appendix L.2 discusses some of the pair-wise contrast results.

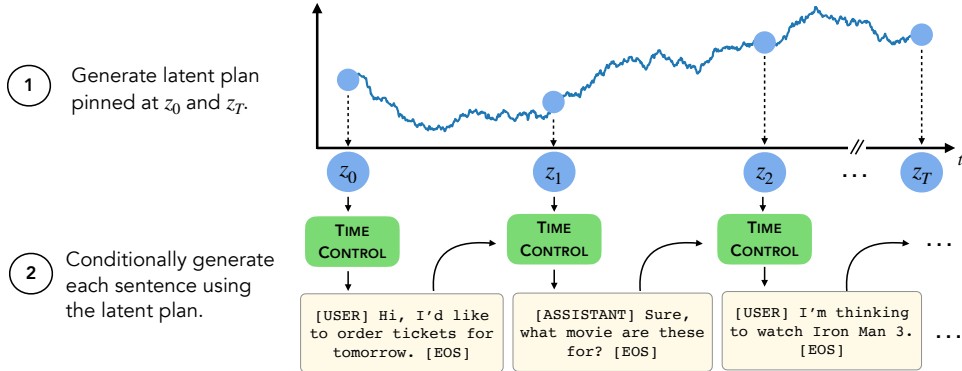

Figure 2: Time Control generates text conditioned on a latent plan. A latent plan is first generated by running Brownian bridge dynamics pinned between a sampled start $z_0$ and goal latent variable $z_T$ forward. A decoder then conditionally generates from this latent plan on a sentence-level.

**RQ1:** *Can Time Control model local text dynamics?* Section 4.1 investigates this question using a sentence ordering prediction task: given two sentences from the same document, we evaluate whether different models can predict their original order.

**RQ2:** *Can Time Control generate locally coherent text?* Section 4.2 investigates this question using the text-infilling task: given prefix and suffix, we evaluate how well different models can fill in between.

**RQ3:** *Can Time Control model global text dynamics?* Section 4.3 investigates this question on text generation for Wikipedia city articles by examining the length of generated sections.

**RQ4:** *Can Time Control generate long coherent documents?* Section 4.4 investigates this question on forced long text generation: we evaluate how well models preserve global text statistics (such as typical section orders and lengths) when forced to extrapolate during generation.

We run Time Control with different latent dimensions ($d = 8, 16, 32$). Our encoder architecture is a frozen, pretrained GPT2 model from Huggingface (Radford et al., 2019; Wolf et al., 2020) and a trainable MLP network. We extract GPT2's last layer hidden state that corresponds to the end-of-sentence (EOS) token and train the 4-layer MLP on top of the hidden state. The MLP network has intermediate ReLU activations and is trained with stochastic gradient descent with a learning rate of 1e-4 and with momentum 0.9.

**Ablations** We perform three ablations on our encoder model. Recall that Time Control **(A) explicitly** models latent structure with **(B) Brownian bridge dynamics** using a **(C) contrastive** loss. **(A)** replaces explicit dynamics with Implicit Dynamics (ID) where future latents are directly predicted with an autoregressive model (van den Oord et al., 2019). **(B)** replaces Brownian bridge dynamics with Brownian motion (BM): latents follow the transition density $z_t|z_s \sim \mathcal{N}(z_s, t-s)$ in Equation 3. Note $z_t$ is centered at $z_s$ and is not conditioned on a goal end-point. **(C)** replaces the contrastive loss with a Variational Auto-Encoder (VAE) and centers the priors over $z_0$ to 0 and $z_T$ to 1, as done in our setup. We use GPT2 as the decoder. Appendix D includes more detail on the ablations.

**Datasets** We use language datasets that elicit different kinds of structure, from section structure to discourse structure to narrative structure. Time Control does not take in any information about the structure, treating each domain the same under its encoding objective. More information and dataset examples are provided in Appendix E. **Wikisection** (Arnold et al., 2019) includes Wikipedia articles on cities split by sections. We adapt this dataset such that each article contains four ordered sections (abstract, history, geography, demographics) marked with section id tokens: Each article is represented as, "[ABSTRACT] text [HISTORY] text [GEOGRAPHY] text [DEMOGRAPHICS] text". **Wikihow** (WH) (Koupaee & Wang, 2018) contains how-to articles organized by a title, method, and steps. We mark each with its own section id tokens: Each article is represented as "[TITLE] text [METHOD] text [STEP] 1 text [STEP] 2 text ..." **Recipe NLG** (Bień et al., 2020) contains recipes, each with a title, ingredients and set of directions. A recipe is constructed as "[TITLE] text

| Method | Wikisection | | TM-2 | | TicketTalk | |
|---|---|---|---|---|---|---|
| | $k = 5$ | $k = 10$ | $k = 5$ | $k = 10$ | $k = 5$ | $k = 10$ |
| GPT2 | $50.3 \pm 5.8$ | $50.2 \pm 6.3$ | $55.7 \pm 5.3$ | $63.6 \pm 7.3$ | $54.7 \pm 6.1$ | $65.0 \pm 8.1$ |
| BERT | $50.9 \pm 4.9$ | $47.8 \pm 9.0$ | $68.8 \pm 3.5$ | $80.7 \pm 3.8$ | $68.4 \pm 5.1$ | $80.4 \pm 6.3$ |
| ALBERT | $49.9 \pm 12.1$ | $49.6 \pm 18.0$ | $\mathbf{81.6 \pm 4.0}$ | $\mathbf{86.1 \pm 7.3}$ | $\mathbf{78.4 \pm 6.7}$ | $\mathbf{89.4 \pm 3.1}$ |
| S-BERT | $50.8 \pm 6.0$ | $48.0 \pm 9.1$ | $73.4 \pm 3.5$ | $\mathbf{83.3 \pm 4.3}$ | $72.1 \pm 5.3$ | $84.2 \pm 5.2$ |
| Sim-CSE | $49.1 \pm 6.4$ | $48.1 \pm 8.5$ | $75.4 \pm 3.8$ | $\mathbf{86.2 \pm 3.9}$ | $\mathbf{75.1 \pm 5.9}$ | $85.2 \pm 3.1$ |
| VAE (8) | $49.5 \pm 5.5$ | $50.5 \pm 5.1$ | $50.5 \pm 4.4$ | $51.5 \pm 6.0$ | $49.9 \pm 1.0$ | $51.2 \pm 1.0$ |
| VAE (16) | $50.1 \pm 5.8$ | $51.3 \pm 4.7$ | $48.8 \pm 4.8$ | $50.8 \pm 4.9$ | $50.1 \pm 1.0$ | $49.5 \pm 1.0$ |
| VAE (32) | $50.5 \pm 5.1$ | $50.0 \pm 6.0$ | $48.0 \pm 5.1$ | $47.3 \pm 5.9$ | $50.0 \pm 1.0$ | $49.3 \pm 1.0$ |
| ID (8) | $49.8 \pm 5.9$ | $50.1 \pm 5.0$ | $60.3 \pm 5.2$ | $65.2 \pm 6.8$ | $59.2 \pm 1.9$ | $66.5 \pm 1.1$ |
| ID (16) | $\mathbf{53.3 \pm 5.4}$ | $55.8 \pm 6.2$ | $60.5 \pm 5.0$ | $67.7 \pm 6.8$ | $60.3 \pm 1.0$ | $68.4 \pm 6.4$ |
| ID (32) | $50.0 \pm 5.0$ | $50.1 \pm 5.0$ | $60.4 \pm 5.3$ | $67.6 \pm 7.1$ | $61.0 \pm 1.0$ | $67.9 \pm 6.5$ |
| BM (8) | $49.8 \pm 5.4$ | $50.0 \pm 5.4$ | $49.8 \pm 5.4$ | $49.9 \pm 5.2$ | $49.7 \pm 5.0$ | $50.6 \pm 5.8$ |
| BM (16) | $50.3 \pm 5.5$ | $50.5 \pm 5.2$ | $49.9 \pm 4.3$ | $51.1 \pm 6.0$ | $50.3 \pm 4.6$ | $50.8 \pm 5.5$ |
| BM (32) | $49.3 \pm 5.6$ | $48.8 \pm 5.8$ | $49.5 \pm 4.7$ | $49.6 \pm 5.2$ | $49.5 \pm 5.6$ | $49.1 \pm 6.1$ |
| TC (8) | $49.23 \pm 5.72$ | $48.3 \pm 6.8$ | $\mathbf{77.6 \pm 7.8}$ | $\mathbf{87.7 \pm 6.9}$ | $71.6 \pm 2.9$ | $82.9 \pm 4.1$ |
| TC (16) | $\mathbf{57.25 \pm 5.30}$ | $\mathbf{65.8 \pm 5.4}$ | $\mathbf{78.2 \pm 8.1}$ | $\mathbf{88.0 \pm 7.1}$ | $71.3 \pm 3.3$ | $82.9 \pm 4.1$ |
| TC (32) | $50.1 \pm 4.8$ | $49.8 \pm 5.8$ | $\mathbf{77.9 \pm 7.9}$ | $\mathbf{87.9 \pm 7.4}$ | $\mathbf{72.0 \pm 3.9}$ | $84.4 \pm 3.9$ |

Table 1: Discourse coherence accuracy measured by the test accuracy of the trained linear classifier, reporting $\mu\pm$ standard error over 3 runs. Random accuracy is 50%. Values are bolded if they are within range of the highest mean score and its corresponding standard error. The highest mean score are marked in gray cells. When applicable, the methods are run with varying latent dimensions marked in parentheses (dim).

[INGREDIENTS] text [DIRECTIONS] text". **Taskmaster-2** (TM-2) (Byrne et al., 2019) contains conversations on finding restaurants between an assistant and a user. The assistant's turn is marked with an "[ASSISTANT]" tag, and the user's turn is marked with a "[USER]" tag. **TicketTalk** (Byrne et al., 2021) contains conversations on booking movie tickets between an assistant and a user. The assistant's and user's turns are similarly marked as in TM-2. **ROC Stories** (Mostafazadeh et al., 2016) is a short 5-sentence stories dataset. No additional tokens are added in this dataset.

## 4.1 Modeling local text dynamics

We evaluate how well Time Control models local text dynamics (**RQ1**) on the discourse coherence setting (Jurafsky, 2000). Discourse coherence is often measured by how well representations capture discourse structure by testing for whether a linear classifier can detect in-order and vs. out-of-order sentence pairs (Chen et al., 2019a). We compare Time Control's encoder against GPT2's last layer's hidden state corresponding to the EOS token (Radford et al., 2019), BERT (Devlin et al., 2019), ALBERT (Lan et al., 2019), Sentence BERT (Reimers et al., 2019), and SimCSE (Gao et al., 2021). The latter 4 methods are designed as sentence embedding models. We also compare to our ablations.

The setup is the following: The encoder takes in two sentences $x_t, x_{t+k}$ to produce their latents $z_t, z_{t+k} \in \mathbb{R}^d$. At random, the latents are fed either in- or out-of-order. A linear classifier is trained on 100 epochs with stochastic gradient descent with a learning rate of 1e-4 and with momentum 0.9. We varied the sentence distance $k \in \{1, 5, 10\}$ and found that on some domains and for $k = 1$, all methods scored near random accuracy; we have omitted those results in the main paper. Otherwise, the results are summarized in Table 1 where we report the mean and standard error accuracy on a held-out discourse dataset of 3000 examples on 3 runs. Our method is able to outperform or compete with sentence embedding specific methods, like ALBERT and SimCSE. Additionally, though GPT2 is used as a base encoder for Time Control, we observe that Time Control greatly improves upon GPT2 with significant gains on TM-2, TicketTalk, and Wikisection (around $7 - 25\%$).

Neither VAE nor BM perform better than random accuracy on any of domains. This suggests that the variational lower bound does not recover the latent embeddings as well as contrastive objectives and the choice of stochastic processes matter for learning informative structural embeddings. ID does best on the conversation datasets, which indicates that implicitly learning text dynamics yields latent representations that can be used for inferring discourse though not as effectively as Time Control.

| Method | BLEU (↑) |
|---|---|
| LM | $1.54 \pm 0.02$ |
| ILM | $3.03 \pm 0.11$ |
| VAE (8) | $0.75 \pm 0.17$ |
| VAE (16) | $0.62 \pm 0.07$ |
| VAE (32) | $0.03 \pm 0.0$ |
| ID (8) | $2.9 \pm 0.3$ |
| ID (16) | $0.9 \pm 0.0$ |
| ID (32) | $1.0 \pm 0.1$ |
| TC (8) | $3.80 \pm 0.06$ |
| TC (16) | $4.30 \pm 0.02$ |
| TC (32) | $\mathbf{5.4 \pm 0.11}$ |

Table 2: BLEU on ground truth infill and generated sentence.

| Method | MM % (↓) |
|---|---|
| GPT2 | $17.5 \pm 0.1$ |
| SD | $10.0 \pm 0.1$ |
| SS | $10.6 \pm 0.1$ |
| VAE (8) | $10.8 \pm 0.1$ |
| VAE (16) | $9.6 \pm 0.1$ |
| VAE (32) | $8.7 \pm 0.1$ |
| ID (8) | $10.8 \pm 0.1$ |
| ID (16) | $154.8 \pm 0.1$ |
| ID (32) | $138.6 \pm 0.1$ |
| BM (8) | $9.2 \pm 0.1$ |
| BM (16) | $17.8 \pm 0.1$ |
| BM (32) | $10.8 \pm 0.1$ |
| TC (8) | $16.8 \pm 0.2$ |
| TC (16) | $\mathbf{7.9 \pm 0.1}$ |
| TC (32) | $9.3 \pm 0.1$ |

Table 3: Percentage of length mismatch (MM) during generation.

| Method | WH | TM-2 | TT |
|---|---|---|---|
| GPT2 | 10.7 | 86.8 | 22.0 |
| VAE (8) | 10.6 | 83.4 | 46.2 |
| VAE (16) | 11.6 | 73.5 | 35.1 |
| VAE (32) | 15.5 | 90.2 | 54.5 |
| ID (8) | 23.1 | 119.1 | 111.1 |
| ID (16) | 38.1 | 87.9 | 55.4 |
| ID (32) | 30.1 | 113.3 | 78.5 |
| BM (8) | 18.1 | 52.0 | 34.9 |
| BM (16) | 12.7 | 44.9 | 75.8 |
| BM (32) | 15.5 | 47.9 | 78.5 |
| TC (8) | **9.6** | 31.1 | 8.0 |
| TC (16) | 15.0 | 9.3 | **5.5** |
| TC (32) | 15.8 | **5.2** | 12.0 |

Table 4: Section lengths deviating from expected length in forced long text generation reported in % (↓).

Noticeably, discourse is more easily inferred on task-oriented conversations like TM-2 and TicketTalk. We hypothesize that the ordering of responses matters more in conversations than enumerative, factual settings like Wikisection. Nonetheless, our model performs above chance on both types of domains. This answers **RQ1** in the positive: Time Control can model local text dynamics, like in conversations and articles.

## 4.2 Generating locally coherent text

We evaluate how well Time Control generates locally coherent text (**RQ2**) on the text-infilling setting. Text-infilling requires a model to take an incomplete text, with missing sentences, and complete it. An example input could be, "Patty was excited about having her friends over. **[blank]** Patty had a great time with her friends." The challenge in performing text infilling is to generate a sentence that is locally coherent with the left and right neighboring sentences.

We follow the task setup in Donahue et al. (2020) where they use the ROCStories dataset. Each story in the dataset contains 5 sentences, one of which is randomly masked. Time Control fills in the masked sentence by running the bridge process pinned at the latent for the prefix, $z_0$, and the latent for the suffix, $z_T$. We compare our method against ILM (Donahue et al., 2020), a state-of-the-art method for text-infilling as well as Donahue et al. (2020)'s LM model, an autoregressive baseline that only sees the prefix. The VAE ablation takes the average of the prefix and suffix embeddings for generation. When using the average, BM is equivalent to Time Control we omit the BM results. ID's autoregressive model predicts the next latent given the prefix for generation.

We evaluate the text coherence with the BLEU score (Papineni et al., 2002), ROUGE (Lin, 2004), BLEURT (Sellam et al., 2020) and BERTScore (Zhang et al., 2019) between the generated and ground truth infill sentence. Due to space constraints, the last three metrics are in the Appendix, Table 17. We also report human evaluations on how coherent the generated sentence is as a fill-in sentence. Participants were asked to rank the generated fill-in sentence from ILM, LM, and Time Control on a scale of 1-5 (not reasonable to very reasonable). Appendix H includes more details on the human experiment setup.

The BLEU scores are summarized in Table 2. Time Control generates fill-in sentences that much more closely overlap with the ground truth than ILM and LM. The ablated methods perform worse to varying degrees. On average, VAE performs worse than LM and ID. This suggests that the interpolating embeddings learned via the variational objective yields embeddings which hurt the autoregressive model's ability to generate locally coherent text.

| Method | Human |
|--------|-------|
| LM | $2.4 \pm 0.06$ |
| ILM | $\mathbf{3.77 \pm 0.07}$ |
| TC (8) | $\mathbf{3.64 \pm 0.07}$ |

Table 6: Human evaluations on text infilling. Scores were ranked between 1 and 5. Higher is better.

| Method | Wikisection | Wikihow | TicketTalk | Recipe |
|--------|-------------|---------|------------|--------|
| GPT2 | $50.4 \pm 1.0$ | $61.5 \pm 3.5$ | $75.8 \pm 1.6$ | $36.8 \pm 3.7$ |
| VAE (8) | $57.1 \pm 3.5$ | $66.3 \pm 1.7$ | $66.1 \pm 4.1$ | $71.4 \pm 0.6$ |
| VAE (16) | $47.3 \pm 3.3$ | $60.5 \pm 2.9$ | $0.8 \pm 0.0$ | $45.6 \pm 0.5$ |
| VAE (32) | $\mathbf{58.3 \pm 0.3}$ | $60.9 \pm 5.6$ | $38.9 \pm 1.7$ | $\mathbf{87.5 \pm 0.2}$ |
| ID (8) | $34.6 \pm 0.0$ | $59.3 \pm 2.9$ | $30.8 \pm 1.6$ | $68.7 \pm 2.3$ |
| ID (16) | $35.7 \pm 0.0$ | $31.7 \pm 0.9$ | $63.9 \pm 3.1$ | $78.1 \pm 3.1$ |
| ID (32) | $47.9 \pm 2.0$ | $16.9 \pm 0.6$ | $45.6 \pm 0.6$ | $85.6 \pm 2.3$ |
| BM (8) | $54.7 \pm 1.9$ | $56.0 \pm 3.5$ | $77.0 \pm 4.7$ | $82.7 \pm 0.4$ |
| BM (16) | $61.4 \pm 1.0$ | $63.8 \pm 4.7$ | $43.3 \pm 2.4$ | $34.9 \pm 1.5$ |
| BM (32) | $44.3 \pm 1.4$ | $50.2 \pm 1.2$ | $14.8 \pm 1.0$ | $87.1 \pm 0.1$ |
| TC (8) | $52.3 \pm 2.5$ | $\mathbf{76.7 \pm 7.5}$ | $81.8 \pm 1.0$ | $41.7 \pm 1.0$ |
| TC (16) | $\mathbf{57.9 \pm 1.0}$ | $63.1 \pm 6.2$ | $\mathbf{88.0 \pm 1.3}$ | $64.1 \pm 1.0$ |
| TC (32) | $36.5 \pm 2.8$ | $59.1 \pm 5.5$ | $83.6 \pm 1.3$ | $76.4 \pm 1.0$ |

Table 5: Ordering in forced long text generation. ROC Stories and TM-2 omitted because they are not applicable.

| Method | Human |
|--------|-------|
| GPT2 | $2.8 \pm 0.06$ |
| TC (8) | $\mathbf{3.6 \pm 0.07}$ |
| TC (16) | $3.4 \pm 0.07$ |
| TC (32) | $3.3 \pm 0.07$ |

Table 7: Human evaluations on tail end quality in forced long text generation. Scores were ranked between 1 and 5. Higher is better.

Human evaluations in Table 6 indicate that Time Control performs competitively with ILM. [4] Upon inspection of the model output, we find that the ablated models struggle to meaningfully decode from the interpolated latent embeddings (Appendix G.1). Oftentimes, the model would start a sentence coherently, but end the sentence by repeating a word over and over again; examples can be found in Appendix G. This suggests the importance of learning a latent space that supports interpolation by construction. All together, the results provide positive evidence for **RQ2**: Time Control can generate locally coherent text due to its well-defined latent dynamics.

### 4.3 MODELING GLOBAL TEXT DYNAMICS

We evaluate how well Time Control models global text dynamics (**RQ3**) by assessing whether the methods mimic document structure on Wikisection. We check whether the generated section lengths match with the average lengths in the dataset. We focus on Wikisection because it is the only dataset with long, well-defined sections (rf. Appendix E on dataset statistics). Each document contains an abstract, history, geography, and demographics section on a city.

Time Control plans a latent trajectory by running the bridge process between the start and end latent, $z_0 \sim p(z_0)$ and $z_T \sim p(z_T)$. We compare Time Control to fine-tuned GPT2. We also include two oracle methods that are fine-tuned GPT2 models with additional section-embedding supervision. One is "sec-dense GPT2" (SD) where each token's embedding contains the current section identity; section $i$'s embedding is added onto the token's positional token embedding. The other is "sec-sparse GPT2" (SS) where the token embedding contains an indicator that is 1 if the token is the start of a new section, and 0 otherwise. For VAE and ID, we calculate the density estimate $p(z_0)$ and $p(z_T)$ and run a linear interpolation between the start and end latents. For BM, we calculate the density estimate $p(z_0)$ and run Brownian motion.

Table 3 reports in percentage how much the generated section lengths deviate from the average section lengths. Time Control best matches the section lengths out of all the methods. One surprising observation was that fine-tuned GPT2 mismatches the section lengths significantly by almost $20\%$. Upon further investigation, we noticed GPT2 overshoots short sections and undershoots long sections; this causes the section length deviations. Adding domain-specific supervision like with SD and SS slightly alleviates this issue. Surprisingly, we find that our method beats the oracle on certain settings, eg. $d = 16$. ID performs extremely poorly in contrast to VAE and BM; this highlights the challenge of interpolating on learned dynamics models which is exacerbated in the long text genera-

---

[4]Note that ILM is in effect an upper bound on model performance as it is trained specifically for text-infilling; TC matching the performance of ILM is a strong result for using latent Brownian bridges for locally coherent generation.

| |
|---|
| **Time Control :** [ USER ] I'm looking for movie tickets please. [ ASSISTANT ] OK, where would you like to see the movie? [ USER ] I'd like to see it at Creek's End, Oregon please. **[...]** [ ASSISTANT ] Is it OK to go ahead and purchase these tickets? [ USER ] Yeah, that would be great. [ ASSISTANT ] OK. I understand that you are interested in tickets for Emma at AMC Mercado 24 tonight at 7:10pm. Is that OK? [ USER ] Yes, please do that. [ ASSISTANT ] OK |
| **GPT2:** [ USER ] Hi! Tonight, I'd like to go to the movies. [ ASSISTANT ] Okay. What theater would you like to go to? [ USER ] Center City. **[...]** [ ASSISTANT ] That will be all for now. Thank you for all your help. N/A [ USER ] Bye Bye. [ ASSISTANT ] N/A [ ASSISTANT ] N/A N/A N/A N/A N/A N/A N/A N/A [ USER ] N/A [ ASSISTANT ] N/A N/A N/A [ USER ] N/A [ ASSISTANT ] N/A N/A N/A [ USER ] N/A N/A N/A N/A N/A N/A N/A N/A N/A N/A [ USER ] N/A N/A N/A N/A **[...]** |

Table 8: Example of forced long text generation on TicketTalk with Time Control vs. fine-tuned GPT2. Both models are forced to extrapolate when generating long texts. They start coherently, but only Time Control extrapolates coherently. For space reasons, some of the text has been removed, marked with **[...]**.

tion settings. The results affirm the importance of Time Control for modeling global text dynamics, such as in matching document structure, thus answering **RQ3** positively.

## 4.4 GENERATING GLOBALLY COHERENT TEXT

We evaluate how well Time Control generates globally coherent text (**RQ4**) where the EOS token is omitted. We refer to this as the forced long text generation setup because the model must extrapolate beyond its natural end point in generation. Appendix C.2 details how the latent plan is generated. For reference, 1000 tokens is about $50\%$ longer than the average Wikisection document (the longest text domain). This setting is intrinsically difficult for auto-regressive models which do not have the ability to "look ahead" during generation (Lin et al., 2021).

We evaluate model extrapolation on three metrics. First is ordering: how well do the models maintain the structure of the document (eg. turn-ordering in TicketTalk)? Second is length consistency. Length consistency captures common failure modes such as a model which stops modeling a conversation between two agents, and instead outputs a long monologue from one agent. Third is human evaluations, which measures the long-tail text quality. We examine the long-tail behavior as the complete 1000-token generation is long.

The results are summarized in Table 4 for length consistency, Table 5 for ordering and Table 7 for human evaluations. Overall, our method maintains text flow the best according to these metrics. A common failure mode of GPT2 is that it produces nonsensical text where it would naturally end generation; this is particularly noticeable on TM-2 and TicketTalk. Figure 8 shows an example of our model's behavior versus GPT2's behavior on generating long texts in TicketTalk. The example illustrates how Time Control continues the dialog whereas GPT2 utters nonsensical text. These results are additionally confirmed by our human evaluation experiments: human evaluators rank Time Control's extrapolation ability better on all latent dimension settings than that of GPT2 (Table 7).

ID performs poorly in forced long text generation, similar to the normal text generation setting: It significantly misorders sections and overshoots section lengths. Although VAE and BM do poorly on length consistency, they perform slightly better than ID on ordering; in Appendix J, we investigate why the VAE baseline performs much better than TC on ordering. This suggests the importance of good latent dynamics for long text generation. These results answer **RQ4** positively: Time Control generates globally coherent text thanks to its ability to plan ahead and correctly generate future latent variables following its latent dynamics.

## 5 CONCLUSION

We propose Time Control, a language model that implicitly plans via a latent stochastic process. Specifically, Time Control looks to Brownian bridge processes as a desirable latent space. Time Control learns to map coherent text to smooth Brownian bridge trajectories. Empirically, we demonstrate this leads to several benefits such as generating more locally and globally coherent text. Although our work focuses on benefits of learning stochastic process latents for language, it can be extended to other domains with sequential data like videos or audio, or extended to handle arbitrary bridge processes without known fixed start and end points.

## 6 REPRODUCIBILITY STATEMENT

In the supplemental, we include a zip file containing our code and processed datasets. We've also included in the appendix how we processed the datasets. Our results on showing equivalence between the triplet classification setting and pairwise classification setting are also included in the appendix.

### ACKNOWLEDGMENTS

REW is supported by the National Science Foundation Graduate Research Fellowship. The authors would give special thanks to CoColab members Mike Wu, Gabriel Poesia, Ali Malik and Alex Tamkin for their support and incredibly helpful discussions. The authors would like to thank the rest of CoCoLab and Rishi Bommasani for reviewing the paper draft. The authors also thank Chris Donahue for helpful pointers on using ILM.

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

## A    SHOWING THE EQUIVALENCE OF THE PAIRWISE AND TRIPLET CLASSIFICATION TASK

In this section, we focus on the classification setup where the goal is to train a classifier to detect in- vs. out-of-order groups of inputs. Specifically, we want to show that the classification setup from Liu et al. (2021) where the inputs come in pairs is equivalent to the setup where the inputs come in triplets. The triplet inputs are what we assume for Time Control.

**Notation**   We denote latent space as $\mathcal{Z}$; we typically do not observe this space directly. We denote the observation space as $\mathcal{X}$; this is the space we observe directly.

We define the Brownian bridge for $t \in [0, T]$ as

$$B_t = B(t) = W(t) - \frac{t}{T}W(T), \tag{4}$$

where $W(t)$ is a standard Wiener process, $\mathcal{N}(0, t)$.

We state the assumptions:

- We assume latents $\{z_t\}_{t \geq 0} \in \mathbb{R}^d$ are drawn from the Brownian bridge process defined by the stochastic differential equation,

$$dz_t = \frac{z_T - z_t}{1 - \frac{t}{T}} dt + dB_t \tag{5}$$

  The intervals at which latents are sampled are every $\Delta t$ step: $z' = z_{t+\Delta t}$.
- We denote the transition probabilities as

$$p_*(z_t | z_0, z_T) := \mathcal{N}\left( (1 - \frac{t}{T})z_0 + \frac{t}{T}z_T, \frac{t(T - t)}{t} \right). \tag{6}$$

  We denote the proposal distribution over possible intermediate triplets $q(z_0', z_t', z_T')$.

Liu et al. (2021) characterize properties of an optimal classifier $h^*(z, z')$ which observes pairs of latents $(z, z')$ and it outputs in $[0, 1]$, a probability indicating whether the pair comes *in order* (ie. $z' = z_t + \Delta t \cdot dz_t$) or *not in order* (ie. a randomly sampled latent). They train $h^*$ using an L2 loss, $\mathcal{L}(h, \{(z, z'), y\})$.

Lemma 1 of their work states the following : The optimum of the contrastive learning objective $\arg\min_h \mathbb{E}_{((z,z'),y)}[\mathcal{L}(h, \{(z, z'), y\})]$ satisfies

$$h^*(z, z') = \frac{p^{\Delta t}(z, z')}{q(z') + p^{\Delta t}(z, z')}. \tag{7}$$

Manipulating this equality, we observe that the transition kernel has the following relation to the classifier which takes in pairs of observations,

$$p^{\Delta t}(z, z') = \frac{q(z')h^*(z, z')}{1 - h^*(z, z')} \tag{8}$$

$$\log p^{\Delta t}(z, z') = \log q(z') + \log h^*(z, z') - \log(1 - h^*(z, z')). \tag{9}$$

Our setting however assumes that the algorithm receives triplets of data points, $z_{t_1}, z_{t_2}, z_{t_3}$. We want to show below that minimizing $\mathcal{L}$ with triplet contrasts in the classification setting still approximates the transition kernel. In particular, we're interested in transitions of a Brownian bridge pinned at $z_0, z_T$: $p^{\Delta t}(z_0, z_t, z_T) = Pr(z_t | z_0, z_T)$.

Let's say we have two positive triplet samples, $(z_{t_1}^i, z_{t_2}^i, z_{t_3}^i)$ and $(z_{t_1}^j, z_{t_2}^j, z_{t_3}^j)$, where $t_1 < t_2 < t_3$. Following Liu et al. (2021), minimizing $\mathcal{L}$ yields the following on each triplet:

$$\log p^{\Delta t}(z_{t_1}^i, z_{t_2}^i, z_{t_3}^i) = \log q(z') + \log h^*(z_{t_1}^i, z_{t_2}^i, z_{t_3}^i) - \log(1 - h^*(z_{t_1}^i, z_{t_2}^i, z_{t_3}^i)) \tag{10}$$

$$\log p^{\Delta t}(z_{t_1}^j, z_{t_2}^j, z_{t_3}^j) = \log q(z') + \log h^*(z_{t_1}^j, z_{t_2}^j, z_{t_3}^j) - \log(1 - h^*(z_{t_1}^j, z_{t_2}^j, z_{t_3}^j)). \tag{11}$$

Taking the difference in log probabilities between Equations 10-11 results in

$$\log p^{\Delta t}(z_{t_1}^i, z_{t_2}^i, z_{t_3}^i) - \log p^{\Delta t}(z_{t_1}^j, z_{t_2}^j, z_{t_3}^j) = \left[ \log h^*(z_{t_1}^i, z_{t_2}^i, z_{t_3}^i) - \log(1 - h^*(z_{t_1}^i, z_{t_2}^i, z_{t_3}^i)) \right]$$
$$- \left[ h^*(z_{t_1}^j, z_{t_2}^j, z_{t_3}^j) - \log(1 - h^*(z_{t_1}^j, z_{t_2}^j, z_{t_3}^j)) \right]. \tag{12}$$

Similar to the pair-wise classification setting, we've shown that minimizing $\mathcal{L}$ in the triplet classification setting results in approximating the transition kernel of the Brownian bridge process.

## B  TRAINING DETAILS

Here we describe the training procedure for the encoder and the fine-tuning procedure for decoding.

**Brownian bridge encoder**   The encoder architecture is a frozen, pretrained GPT2 model from Huggingface (Radford et al., 2019; Wolf et al., 2020) and a trainable MLP network. We extract the GPT2's last layer hidden state that corresponds to the end-of-sentence (EOS) token and train the 4-layer MLP on top of the hidden state. The MLP network has intermediate ReLU activations and is trained with stochastic gradient descent with a learning rate of 1e-4 and with momentum 0.9. We train the encoder for 100 epochs on each of the datasets.

The text fed into GPT2 are fed in on a sentence level. This means that the input $x_t$ refers to the $t$'th sentence of a document. The sentences are separated from each other in the main text as " . " which is added to the tokenizer as a separate token for indexing convenience.

**Fine-tuning GPT2 with latent embeddings**   After training the encoder, we run it on the training dataset to collect an accompanying latent trajectory for each text. The encoder is run on the dataset at a sentence level: we separate the text by sentences and pass the sentences through the encoder.

The sentence latent embeddings are aligned with the tokens of that sentence and offset by one token before the start of that sentence token. Let's illustrate by an example. We denote [SOS] as the start of the document token, [s1] as sentence 1 tokens and [s2] as sentence 2 tokens. [ . ] is the period token which we've added into the tokenizer. $z_i$ denote the latent variable corresponding to the $i$'th sentence.

Let's say that the sequence fed into GPT2 is "[SOS] [s1] [s1] [s1] [ . ] [s2] [s2] [s2]". Then the corresponding latent trajectory is "$z_1, z_1, z_1, z_1, z_2, z_2, z_2, z_2$". The latent variables are added onto the positional embeddings. We then fine-tune GPT2 as normal.

## C  GENERATION WITH EMBEDDINGS

### C.1  NORMAL TEXT GENERATION

We first sample a start and end latent, $z_0 \sim p(z_0), z_T \sim p(z_T)$ where $p(z_0), p(z_T)$ are calculated as the density estimates over the training dataset. We pin our trajectory to the start and end latent, and run the Brownian bridge using Equation **??**. For normal long text generation, we set $T$ to be the average number of sentences in each of the dataset. For forced long text generation, we set $T$ to be proportional to the number of sentences needed in order to generate 1000 tokens. By the end, we have a trajectory $z_0, z_1, ..., z_T$.

Generation starts with feeding the SOS token and the first latent $z_0$. Once GPT2 emits a [ . ] token and terminates sentence $t$, we transition to the next latent $z_{t+1}$. This process continues until GPT2 is finished with generation. If GPT2 generates more sentences than there are latents in the trajectory, the last latent $z_T$ is used until the end of generation.

### C.2  FORCED LONG TEXT GENERATION

Let $S_{\mathrm{avg}(\mathcal{D})}$ denote the average number of sentences in a document and $T_{\mathrm{avg}(\mathcal{D})}$ denote the average number of tokens in a document. Rather than planning a trajectory of length $S_{\mathrm{avg}(\mathcal{D})}$ (the average number of sentences in a document) which is what is done in Section 4.3 for normal text generation, we scale the trajectory length to $c \cdot S_{\mathrm{avg}(\mathcal{D})}$. $c$ is determined by how many more tokens we need in order to fill up to GPT-2 maximum context length of 1024: $c = \frac{1024 - T_{\mathrm{avg}(\mathcal{D})}}{T_{\mathrm{avg}(\mathcal{D})}}$.

## D  ABLATIONS

The following methods ablate the encoder model in Time Control; in other words, the ablations dissect the assumptions we make in the first step of Time Control described in Section 3.1. Recall that Time Control **(A) explicitly** models latent structure with **(B) Brownian bridge dynamics** using

a **(C) contrastive** loss. **(A)** replaces explicit dynamics with Implicit Dynamics (ID) where future latents are directly predicted with an autoregressive model (van den Oord et al., 2019). **(B)** replaces Brownian bridge dynamics with Brownian motion (BM) which doesn't rely on pinning trajectories. Latents follow the transition density $z_t|z_s \sim \mathcal{N}(z_s, t - s)$. **(C)** replaces the contrastive loss with the Variational Auto-Encoder (VAE). Below we detail how these ablations are implemented.

## D.1 IMPLICIT DYNAMICS (ID)

The Implicit Dynamics ablation is where we compare our *explicit* dynamics objective to van den Oord et al. (2019) which suggests an *implicit* latent dynamics objective. This ablation thus changes the learned latent dynamics. In van den Oord et al. (2019), they train two models. One is a non-linear encoder $g_{enc}(x_t) = z_t$ which takes observation $x_t$ (eg. a sentence) and maps it to a latent representation $z_t$. Two is an autoregressive context model $g_{ar}(z_{\leq t}) = c_t$ which summarizes a sequence of latent variables $z_{\leq t}$ into a context latent representation $c_t$. Rather than directly predicting a future observation $x_{t+k}$ ($k$ steps from the current timestep $t$), they model the density ratio that preserves the mutual information between $x_{t+k}$ and the context variable $c_t$:

$$f_k(x_{t+k}) \propto \frac{p(x_{t+k}|c_t)}{p(x_{t+k})} \tag{13}$$

They model this with a log-bilinear model $f_k(x_{t+k}) = \exp(z_{t+k}^T W_k c_t)$ which applies a linear transformation $W_k$ for modelling latents k-steps away from timestep $t$. This way, they avoid directly learning the generative model $p(x_{t+k}|c_t)$.

They train both models jointly via a contrastive InfoNCE loss. Given a set $X = \{x_1, \ldots, x_N\}$ of $N$ random samples (eg. sentences from different documents) containing one positive sample from $p(x_{t+k}|c_t)$ and $N - 1$ negative samples from a proposal distribution $p(x_{t+k})$, they optimize:

$$\mathcal{L} = -\mathbb{E}_X \left[ \log \frac{f_k(x_{t+k}, c_t)}{\sum_{x_j \in X} f_k(x_j, c_t)} \right]. \tag{14}$$

The encoder $g_{enc}$ for Implicit Dynamics has the same architecture as Time Control. The context encoder $g_{ar}$ using a 2400-hidden-unit GRU, as done in van den Oord et al. (2019). We then train both encoders using the InfoNCE loss, as done in prior work. Since $g_{enc}$ and $g_{ar}$ are trained to align latents up to a linear rotation, we use $g_{enc}$ for extracting the sentence embeddings.

## D.2 BROWNIAN MOTION (BM)

The Brownian motion ablation is where we remove goal-conditioning in the dynamics. The main change is in distance function in Equation 3. BM instead optimizes the following equation:

$$\mathrm{d}(x_t, x_{t'}; f_\theta) = -\frac{1}{2\sigma^2} \| \underbrace{f_\theta(x_{t'})}_{z_{t'}} - \underbrace{f_\theta(x_t)}_{z_t} \|_2^2 \tag{15}$$

where $\sigma^2$ is the variance in of the Wiener process, $\sigma^2 = t' - t$.

## D.3 VARIATIONAL AUTO-ENCODER (VAE)

The Variational Auto-Encoder ablation is where we compare our contrastive objective with a variational one. This ablation changes the encoder objective from Section 3.1 from Equation 2 to the ELBO objective. Below we derive the ELBO objective. Similar to the contrastive objective, we're deriving the ELBO over the triplet dynamics in the Brownian bridge.

$$\log p(\mathbf{x}) \geq \mathbb{E}_{q_\phi(\mathbf{z}|\mathbf{x})}[\log \frac{p(\mathbf{x}, \mathbf{z})}{q(\mathbf{z}|\mathbf{x})}]$$

$$= \mathbb{E}_{q_\phi(\mathbf{z}|\mathbf{x})}[\log \frac{p(x_0, x_t, x_T, z_0, z_t, z_T)}{q(z_0, z_t, z_T|x_0, x_t, x_T)}]$$

$$= \mathbb{E}_{q_\phi(\mathbf{z}|\mathbf{x})}[\log \frac{p(x_0|z_0)p(x_t|z_t)p(x_T|z_T)p(z_t|z_0, z_T)p(z_0)p(z_T)}{q(z_t|z_0, z_T, x_t)q(z_0|x_0)q(z_T|x_T)}]$$

$$= \mathbb{E}_{q_\phi(\mathbf{z}|\mathbf{x})}[p(x_0|z_0)p(x_t|z_t)p(x_T|z_T)] + \mathbb{E}_{q_\phi(\mathbf{z}|\mathbf{x})}[\log \frac{p(z_t|z_0, z_T)}{q(z_t|z_0, z_T, x_t)}]$$

$$+ \mathbb{E}_{q_\phi(\mathbf{z}|\mathbf{x})}[\log \frac{p(z_0)}{q(z_0|x_0)}] + \mathbb{E}_{q_\phi(\mathbf{z}|\mathbf{x})}[\log \frac{p(z_T)}{q(z_T|x_T)}]$$

$$= \mathbb{E}_{q_\phi(\mathbf{z}|\mathbf{x})}[\log p(x_0|z_0)p(x_t|z_t)p(x_T|z_T)]$$
$$- D_{\text{KL}}(q(z_t|z_0, z_T, x_t)\|p(z_t|z_0, z_T)) - D_{\text{KL}}(q(z_0|x_0)\|p(z_0)) - D_{\text{KL}}(q(z_T|x_T)\|p(z_T))$$

We assume the priors over $z_0$ is 0-centered and $z_T$ is 1-centered, which is similar to our Brownian Bridge setup. The encoder $q_\phi(z|x)$ is parameterized with the same architecture as our encoder. The decoder $p(x|z)$ is a fine-tuned GPT2 model.

## E    DATASET INFORMATION

For each dataset, text examples were filtered out if they did not fit within GPT2's context length of 1024 tokens. We also added the token " . " for each setting to mark the end of a sentence. This was done for indexing purposes, eg. when aligning the latent embeddings.

**Wikisection**    (Arnold et al., 2019) includes Wikipedia articles on cities split by sections. We adapt this dataset such that each article contains four ordered sections (abstract, history, geography, demographics) marked with section id tokens: Each article is represented as, "[ABSTRACT] text [HISTORY] text [GEOGRAPHY] text [DEMOGRAPHICS] text". These section id tokens are added to the tokenizer.

The training dataset contains 1420 articles. The section lengths have the following breakdown measured in the number of BPE tokens (GPT2 tokenizer):

- Abstract: $75.8 \pm 1.4$
- History: $191.5 \pm 3.7$
- Geography: $83.9 \pm 1.5$
- Demographics: $342.6 \pm 4.6$

The test dataset contains 431 articles. The section lengths have a similar breakdown:

- Abstract: $73.5 \pm 2.6$
- History: $180.2 \pm 6.2$
- Geography: $85.2 \pm 2.7$
- Demographics: $332.5 \pm 8.6$

The ordering metric used in Table 5 is 1 if all four section ids occur exactly once and come in the order as they are listed above.

The length mismatch in % used in Table 3 is calculated with respect to the training set lengths.

**Wikihow**    (Koupaee & Wang, 2018) contains how-to articles organized by a title, method, and steps. Each article includes multiple methods for completing a multi-step procedural task such as *"How to Register to Vote"*. We scraped all the available English articles covering a wide range of

topics following Koupaee & Wang (2018). We mark each with its own section id tokens: Each article is represented as "[TITLE] text [METHOD] text [STEP] 1 text [STEP] 2 text ..."

The training dataset contains 1566 articles. The section lengths are,

- Title: $10.4 \pm 2.2$
- Method: $8.7 \pm 2.2$
- Steps (total step length): $480.2 \pm 231.5$

The test dataset contains 243 articles. The section lengths are,

- Title: $10.7 \pm 2.2$
- Method: $8.6 \pm 2.1$
- Steps (total step length): $480.1 \pm 224.0$

The ordering metric used in Table 5 is 1 if all the section ids appear in order, the TITLE and METHOD section ids are not repeated, and the step numbers come in order. It's 0 otherwise.

The deviation in section length measured in Table 4 is calculated with respect to the training set lengths. We check for whether the models are able to maintain the section lengths when it has to extrapolate. The most common failure mode is that the model generates incoherent text and allocates this text to the last section of what it's generated thus far, resulting in the deviation in section lengths.

**Recipe NLG** (Bień et al., 2020) contains recipes, each with a title, ingredients and set of instructions. A recipe is constructed as "[TITLE] text [INGREDIENTS] text [DIRECTIONS] text".

The training dataset contains 4000 recipes. The section lengths are,

- Title: $9.7 \pm 3.4$
- Ingredients: $23.8 \pm 4.5$
- Directions (total step length): $62.0 \pm 14.5$

The test dataset contains 1000 recipes. The section lengths are,

- Title: $9.4 \pm 3.0$
- Ingredients: $24.1 \pm 4.5$
- Directions (total step length): $63.3 \pm 13.7$

The ordering metric used in Table 5 is 1 if all the section ids appear exactly once and in order. It's 0 otherwise.

**Taskmaster-2** (TM-2) (Byrne et al., 2019) which contains conversations on finding restaurants between an assistant and a user. The assistant's turn is marked with an "[ASSISTANT]" tag, and the user's turn is marked with a "[USER]" tag.

The training dataset contains 2000 conversations. The section lengths are,

- User: $11.8 \pm 5.6$
- Assistant: $18.0 \pm 3.7$

The test dataset contains 1276 conversations. The section lengths are,

- User: $11.7 \pm 5.6$
- Assistant: $19.5 \pm 3.0$

The ordering metric used in Table 5 does not apply here because the user and assistant don't take turns in the dialog.

| Method | Wikisection | Wikihow | TM-2 | TicketTalk | Recipe |
|--------|-------------|---------|------|------------|--------|
| GPT2 | 5.9 | 15.3 | 4.5 | 4.4 | 7.5 |
| sec-dense | 5.9 | – | – | – | – |
| sec-sparse | 5.9 | – | – | – | – |
| VAE (8) | 5.5 | 15.3 | 4.5 | 4.1 | 7.3 |
| VAE (16) | 5.5 | 15.3 | 4.5 | 4.1 | 7.3 |
| VAE (32) | 5.5 | 15.3 | 4.5 | 4.1 | 7.3 |
| ID (8) | 5.5 | 14.9 | 4.5 | 4.0 | 7.3 |
| ID (16) | 5.4 | 14.9 | 4.3 | 3.9 | 7.0 |
| ID (32) | 5.3 | 14.9 | 4.2 | 3.9 | 6.7 |
| BM (8) | 5.5 | 15.2 | 4.5 | 4.1 | 7.3 |
| BM (16) | 5.5 | 15.3 | 4.5 | 4.1 | 7.3 |
| BM (32) | 5.5 | 15.2 | 4.5 | 4.1 | 7.3 |
| TC (8) | 5.5 | 15.2 | 4.3 | 4.0 | 6.9 |
| TC (16) | 5.5 | 15.3 | 4.3 | 4.0 | 6.9 |
| TC (32) | 5.5 | 15.2 | 4.3 | 4.0 | 7.0 |

Table 9: Perplexity after fine-tuning.

The deviation in section length measured in Table 4 is calculated with respect to the training set lengths. We check for whether the models are able to maintain the utterance lengths between the user and assistant when it has to extrapolate. The most common failure mode is that the model generates incoherent text and allocates this text to the last section of what it's generated thus far, resulting in the deviation in section lengths.

**TicketTalk** (Byrne et al., 2021) which contains conversations on booking movie tickets between an assistant and a user. The assistant's and user's turned are similarly marked as in TM-2.

The training dataset contains 2000 conversations. The section lengths are,

- User: $11.8 \pm 5.6$
- Assistant: $18.0 \pm 3.7$

The test dataset contains 1276 conversations. The section lengths are,

- User: $11.7 \pm 5.6$
- Assistant: $19.5 \pm 3.0$

The deviation in section length measured in Table 4 is calculated similarly to TM-2.

The ordering metric used in Table 5 applies because the user and assistant take turns in the dialog. The ordering metric is 1 if the user and assistant take turns in the conversation. It's 0 otherwise.

**ROC Stories** (Mostafazadeh et al., 2016) is a short stories dataset. Each story contains 5 sentences. No additional tokens are added in this dataset. The training dataset contains 2000 stories, and the test dataset contains 1000 stories.

# F  PERPLEXITY AFTER FINE-TUNING

Table 9 reports the final perplexity scores after fine-tuning GPT2 on the different domains with the methods. We fine-tune for 10 epochs and checkpoint the models every 1000 steps; we keep the model checkpoint that scores the lowest PPL on a held-out validation set.

## G  GENERATION EXAMPLES

We provide example outputs from the different models on text infilling. We denote the prefix sentences as "LEFT CONTEXT", the suffix sentences as "RIGHT SENTENCES", the generated output from the models as "GENERATED" and the ground truth infill sentence as "GT".

### G.1  TEXT INFILLING

**LM**  See Table 10.

Table 10: Example of text infilling with Donahue et al. (2020)'s **LM** model on ROCStories.

| |
|---|
| LEFT CONTEXT: John couldn't wash his clothes because his washing machine was broken. |
| GENERATED: He had an idea. |
| GT: John didn't know how to fix washing machines. |
| RIGHT CONTEXT: He bought a book to tell him how to repair it. He changed the fan belt. When he plugged it back in, it worked. |

| |
|---|
| LEFT CONTEXT: I found a stray cat yesterday. I gave it some milk. It came back again today. I was going to give it some milk but didn't. |
| GENERATED: I guess I need to get something. |
| GT: I had run out of milk. |
| RIGHT CONTEXT: |

| |
|---|
| LEFT CONTEXT: Toni took a trip to the local fair with her family. She immediately laid eyes on a pretty balloon. The balloon was red and star shaped. Unfortunately it slipped out of her hands. |
| GENERATED: Her swollen fingers had horrible rashes. |
| GT: She was sad to see it go. |
| RIGHT CONTEXT: |

| |
|---|
| LEFT CONTEXT: Miles was a roofer. He loved his job and was very good at it. |
| GENERATED: He dreamed of finding a good title. |
| GT: However, one day he was not paying attention and fell off the roof. |
| RIGHT CONTEXT: Luckily, he only broke his leg. Miles decided that he needed to be more careful. |

| |
|---|
| LEFT CONTEXT: Lindsey was feeling especially antsy at school. |
| GENERATED: The school wasn't paying much attention to her. |
| GT: She did not want to go to Lacrosse practice after school. |
| RIGHT CONTEXT: She tried to fake a sickness, but nobody believed her. Lindsey ended up skipping practice without an excuse. She hoped she wouldn't get in trouble. |

| |
|---|
| LEFT CONTEXT: Last night Jack went on a blind date. When he arrived at the restaurant a he heard someone yell his name. He spotted a blonde waving from across the room. |
| GENERATED: She was a waiter. |
| GT: She was nothing like the photo she sent. |
| RIGHT CONTEXT: But he figured he would try to make the most of it. |

| |
|---|
| LEFT CONTEXT: Rico was working really hard on his essay for English. After working on it all night, he decided to take a short nap. He slept in and was late for for school. When he got to school he noticed that his essay was gone. |
| GENERATED: teacher found the essay in his bag and he passed it. |
| GT: Rico's teacher let him turn in the essay the next day. |
| RIGHT CONTEXT: |

**ILM**  See Table 11.

Table 11: Example of text infilling with Donahue et al. (2020)'s **ILM** model on ROCStories.

---

LEFT CONTEXT:
GENERATED: My 98 year old friend and I played blackjack yesterday.
GT: Last week's family game night was intense.
RIGHT CONTEXT: We were playing Monopoly and nobody made any headway for hours. Everyone was trying their hardest to win and the game kept going. It wasn't until we finally decided to check the rules that we knew why. There were many different pieces missing.

---

LEFT CONTEXT: Tom was jealous of his brother. His brother was older and stronger. His brother went wherever he wanted.
GENERATED: Tom decided to use steroids.
GT: One day his brother was grounded for staying out too late.
RIGHT CONTEXT: This made Tom really happy.

---

LEFT CONTEXT: His first time at the opera. He only went after his girlfriend begged. He sat for three hours in sheer boredom. Finally it was over.
GENERATED: He turned on the tv and heard her thanking him.
GT: He learned he didn't like the opera.
RIGHT CONTEXT:

---

LEFT CONTEXT: My best friend played a video game online. One day, she met a nice boy there. They talked every day and fell in love. They finally met in person and hit it off.
GENERATED: Now, they are best friends.
GT: The two became a very loving couple.
RIGHT CONTEXT:

---

LEFT CONTEXT: The family decided to go to the park for Veteran's Day.
GENERATED: They asked permission from the park manager.
GT: We had a barbecue outside.
RIGHT CONTEXT: In addition, everyone decided to take a swim in the water. Before we left, we had a huge cake. We then swam for an hour at the lake.

---

LEFT CONTEXT: Jennifer always wanted to have the birthday of her dreams. Her parents surprised her with a big birthday party at the park. All of her friends and family came to celebrate her special day. She was so happy as she blew out the candles on her cake.
GENERATED: Her parents had made a big surprise to her by turning the cake into a carnival.
GT: It was the best day of birthday she could have imagined.
RIGHT CONTEXT:

---

LEFT CONTEXT: Terry's daughter was in extreme pain in her mouth.
GENERATED: Terry and her husband decided to stop the baby from teething.
GT: After an evaluation, Terry realized that her daughter was teething.
RIGHT CONTEXT: Once Terry realized she was teething, Terry administered medication. After the medication, Terry placed teething gel on the baby's teeth. Terry got the baby to calm down.

---

**Time Control**   See Table 12.

Table 12: Example of text infilling with **Time Control** (d=8) on ROCStories.

---

LEFT CONTEXT:My day on Sunday this week went as expected. My family and I attended Church from 8 to 11 in the morning.
GENERATED: We went to a local church in the afternoon .
GT: Afterwards, we went to eat dinner at a restaurant.
RIGHT CONTEXT: The rest of my day was spent doing homework assignments. I then went to sleep feeling happy about my life.

---

LEFT CONTEXT: His first time at the opera. He only went after his girlfriend begged. He sat for three hours in sheer boredom. Finally it was over.
GENERATED: He was so happy he didn't want to leave.
GT: He learned he didn't like the opera.
RIGHT CONTEXT:

---

LEFT CONTEXT: My best friend played a video game online. One day, she met a nice boy there. They talked every day and fell in love. They finally met in person and hit it off.
GENERATED: Until that day, they were married.
GT: The two became a very loving couple.
RIGHT CONTEXT:

---

LEFT CONTEXT: The other day at the clinic I had to help put a dog down. He seemed really sad and lonely. Like he knew what was going to happen.
GENERATED: He was going to die soon .
GT: As we laid it down and it took its final breaths it stared at me.
RIGHT CONTEXT: I stayed calm, but cried after we were finished.

---

LEFT CONTEXT: Tom was jealous of his brother. His brother was older and stronger. His brother went wherever he wanted.
GENERATED: Tom was jealous of his brother.
GT: One day his brother was grounded for staying out too late.
RIGHT CONTEXT:

---

LEFT CONTEXT: Jays habit of buying expensive beer was catching up to him. He was spending more money on beer than food. He needed to find another source of income to support this habit. A friend recommended he try out Amazon MTurk.
GENERATED: He found the site and bought a few beers.
GT: Jay become slightly less poor from Amazon Mturk.
RIGHT CONTEXT:

---

LEFT CONTEXT: John couldn't wash his clothes because his washing machine was broken.
GENERATED: John went to the store to buy a new one .
GT: John didn't know how to fix washing machines.
RIGHT CONTEXT: He bought a book to tell him how to repair it. He changed the fan belt. When he plugged it back in, it worked.

---

Table 13: Example of text infilling with **Time Control** (d=16) on ROCStories.

LEFT CONTEXT: Tom was jealous of his brother. His brother was older and stronger. His brother went wherever he wanted.
GENERATED: Tom's brother was very jealous of his brother.
GT: One day his brother was grounded for staying out too late.
RIGHT CONTEXT: This made Tom really happy.

LEFT CONTEXT: Jackie was 11 and had to get braces. She was worried about what her friends would think. She tried to hide them when she first got them.
GENERATED: But she was too embarrassed to tell them .
GT: Eventually her friends saw them and she was embarrassed.
RIGHT CONTEXT: Her friends noticed she was embarrassed and decided to comfort her.

LEFT CONTEXT: Sally was going to surprise the office with a cake. Sally felt that a cake would be a good way to make them smile. She went to the supermarket to pick up the cake.
GENERATED: She bought the cake and was very happy with it.
GT: At the office she gathered the employees around the conference table.
RIGHT CONTEXT: She then brought out a cake and they all felt better.

LEFT CONTEXT: Lars was playing XBOX. His controller stopped working during a game. Lars didn't have a car so he had to walk all the way to the store. The store was being remodeled when he got there so he went to another.
GENERATED: Video games were all over the floor.
GT: Lars wasn't able to find a controller.
RIGHT CONTEXT:

LEFT CONTEXT: Emma had been working as a dishwasher. Her hands cracked and bled from the hot soapy water. Then her mom noticed and concocted a special salve for her. Emma used the salve every night before bed.
GENERATED: Soon she was able to keep her hands clean and happy.
GT: Her hands got better.
RIGHT CONTEXT:

LEFT CONTEXT: Jerome dribbled the ball quickly. Sam tried to grab it from him, but wasn't fast enough. He chased Jerome down the court.
GENERATED: He scored a point with a long shot .
GT: Jerome pushed Sam backwards and threw the ball.
RIGHT CONTEXT: Jerome scored points for his team.

LEFT CONTEXT: Class started in 10 minutes and I had a math assignment due. My older friend volunteered to do it for me. Unfortunately, my teacher found out about the copying.
GENERATED: She was very upset with me for doing the assignment.
GT: She crumbled my paper and threw it in the trash.
RIGHT CONTEXT: My teacher gave me an F for the assignment.

Table 14: Example of text infilling with **Time Control** (d=32) on ROCStories.

---

LEFT CONTEXT: I went to the park to play frisbee with my dog.
GENERATED: We played all day .
GT: I tossed the frisbee to my dog and he would catch it in his mouth.
RIGHT CONTEXT: I accidentally threw the frisbee too far. The frisbee landed into the pond. But my dog went and got it.

---

LEFT CONTEXT: I was tired of working at Walmart. The hours were bad. The store music was awful.
GENERATED: I was very bored .
GT: I handed my two weeks in to the manager.
RIGHT CONTEXT: I then found another job and was happy.

---

LEFT CONTEXT: Sam bought a new SUV. It was all wheel drive. He figured he would take it off road. He hit a few hard bumps and broke his suspension.
GENERATED: Unfortunately he had to pay a lot of money for it.
GT: Sheepishly, he brought it to the dealership for repair.
RIGHT CONTEXT:

---

LEFT CONTEXT: Missy got drunk and went to get a tattoo. She decided to get a tattoo on her forehead. The next day, Missy was horrified at what she had done.
GENERATED: Her tattoo was on her forehead!
GT: Missy scraped up her money to pay for a tattoo removal procedure.
RIGHT CONTEXT: After much wasted money, the tattoo was gone.

---

LEFT CONTEXT: Jake was going on a road trip to see his family. He Got in the car and drove.
GENERATED: Jake's family was driving down the road .
GT: The car's tires exploded due to too much air.
RIGHT CONTEXT: Jake hitchhiked for 30 miles. When Jake got to his family he was happy his trip was over.

---

LEFT CONTEXT: Bob decided to start a business.
GENERATED: However, he did not know the market at all .
GT: He opened up a grocery store and was doing very well.
RIGHT CONTEXT: After a year, his profits dropped and he had to declare bankruptcy. Bob was sad to see his business fail. Bob worked hard and reopened his business.

---

LEFT CONTEXT: I hadn't seen my girlfriend in a while. She got a new job so it's hard to talk. The job takes up all of her time.
GENERATED: I had to ask her to go out with me .
GT: Finally she called me to hang out.
RIGHT CONTEXT: I was really happy to see her and we made plans.

---

**Variational Auto-Encoder**   See Table 15.

Table 15: Example of text infilling with **Variational Auto-Encoder** on ROCStories.

---

LEFT CONTEXT: I went to the park to play frisbee with my dog.
GENERATED: served served served **[...]**
GT: I tossed the frisbee to my dog and he would catch it in his mouth.
RIGHT CONTEXT: I accidentally threw the frisbee too far. The frisbee landed into the pond. But my dog went and got it.

---

LEFT CONTEXT: Tim had a dentist appointment today. He was going to get his wisdom teeth pulled.
GENERATED: . .
GT: His dentist numbed his gums.
RIGHT CONTEXT: His wisdom teeth popped out instantly. It was less hassle than he thought it would be.

---

LEFT CONTEXT: Tom was studying for the big test. He then fell asleep do to boredom.
GENERATED: The The The The The The The The The **[...]**
GT: He slept for five hours.
RIGHT CONTEXT: He woke up shocked. Tom hurried to study as much as possible before the test.

---

**Implicit Dynamics**    See Table 16.

Table 16: Example of text infilling with **Implicit Dynamics** on ROCStories.

---

LEFT CONTEXT: Tom was jealous of his brother. His brother was older and stronger. His brother went wherever he wanted.
GENERATED: he wasn't like him. he wasn't like him............................................................ isn't... doesn't... doesn't.. don't. don't. don't. don't. don't. don't. don't. don't. don't. don't. don't. don't. don't. don't. don't. don't. don't. don don't. don don't don't. don don't don't don't don't don't don't don't don don don don **[...]**
GT: One day his brother was grounded for staying out too late.
RIGHT CONTEXT: This made Tom really happy.

---

LEFT CONTEXT: The family decided to go to the park for Veteran's Day.
GENERATED: I didn't I knew why why why why **[...]**
GT: We had a barbecue outside.
RIGHT CONTEXT: In addition, everyone decided to take a swim in the water. Before we left, we had a huge cake. We then swam for an hour at the lake.

---

## G.2    FORCED LONG TEXT GENERATION

We include examples of GPT2 long text generation in Figures 3 and 4.

## H    HUMAN EXPERIMENTAL SETUP

We use Mechanical Turk to collect human evaluation for long text generation and text infilling models. Examples from all the systems are selected randomly and each example is evaluated by 10 annotators.

### H.1    LONG TEXT GENERATION EVALUATION

The annotators are provided with a dialogue between a movie assistant and a user generated by GPT and Time Control, where the goal of the assistant is to help the user to book a movie ticket. We show the last few generated interactions between the assistant and the user to the annotators, and ask them to assess whether the provided snippet could be a reasonable ending for such a conversation. We use a 5-point Likert scale where 1 corresponds to a "Very Unreasonable Ending" and 5 corresponds to a "Very Reasonable Ending".

```
[ USER ] I'm interested in buying some tickets for a movie tonight. [
ASSISTANT ] OK. Can you tell me the city and state where the theater is
located? [ USER ] It's been so long since we've been to the movies that I want
to get out. It can't get any better. [ ASSISTANT ] Great. Can you tell me the
city and state where you plan to see the movie? [ USER ] I'll be in Santa
Barbara, California at the moment. [ ASSISTANT ] Got it. Are you interested in
a particular movie? [ USER ] There's a couple of nice theaters in town. [
ASSISTANT ] OK. Is there a movie you have in mind? [ USER ] I'm thinking
action. [ ASSISTANT ] Ok. Let's see what action films are out in that area. [
ASSISTANT ] I found one action movie at AMC Mountain 16: No Time To Die. Are
you interested in that one? [ USER ] My best friend Sally has been wanting to
see that one, and I guess we'll go with that one. [ ASSISTANT ] OK. Are you
interested in this movie though? [ USER ] Definitely interested in seeing No
Time to Die. [ ASSISTANT ] OK. Remaining showtimes for No Time To Die at AMC
Mountain 16 are 4:30pm, 6:40pm and 9:10pm. [ USER ] 9:10 pm is good. [
ASSISTANT ] Got it. And how many tickets? [ USER ] Just me and a couple other
guys. [ ASSISTANT ] All right. Let me confirm that you'd like three tickets
for No Time To Die at AMC Mountain 16 tonight at 9:10pm. Is that all correct?
[ USER ] Yes that's right. [ ASSISTANT ] Is it OK to go ahead and purchase
these tickets? [ USER ] Yes go right ahead. [ ASSISTANT ] OK. Your tickets are
purchased and details for how to proceed have been sent to your email address.
Can I help with anything else? [ USER ] No. [ ASSISTANT ] OK. Enjoy your
movie! As always, have a good day. javascript ive been! havn bout ive heard
it. dont think to much. in there. lmao. id appreciate it. reccommended. The
email address for your tickets is theater. There's no problem at all with
anything else. thanks! sign off. Your tickets are confirmed and they've been
sent to your email address. thanks a lot. your welcome. enjoy your movie.
-1bye. I'm very, very sorry for leaving you behind. Thanks for calling. Is
there anything else I can help with? dont remember the last time we went out.
I'm very sorry. Thank you very much. arn. Thanks for all of your help. Enjoy
the movie. wwww, bye. -en Chinese girlfriend, bye. can you enjoy your movie.
Bye. remaining showtimes for No Time To Die at AMC Mountain 16 are 4:30pm,
6:40pm and 9:10pm. This is all for right now. N/A Thanks for all of your
assistance. @ @ b/r/r/r/r abys mam. @ cornyyyyyyyyyyy. BULLY BULLY MOVIE FOR
YOU! Thank you for calling. Enjoy your movie! Your movie. Aww, bye. happy
movie. thanks for calling. -en . Thanks for using our services. > .
```

Figure 3: Example 1 of GPT2 forced long text generation.

## H.2 TEXT INFILLING EVALUATION

Given the preceding and following context, we ask annotators to evaluate whether the generated fill-in sentence is a reasonable completion. We evaluate generated fill-in sentence by ILM, LM and Time Control on a scale of 1-5.

## I EXAMPLE TRAJECTORIES OF THE LEARNED LATENT SPACE

See Figures 7-10 for example latent trajectories over coherent vs. incoherent (randomly scrambled) held-out Wikisection documents. These trajectories show the recovered latent structure by Time Control and the three ablations from our work. Figure 7 are latent trajectories from Time Control. Figure 8 are latent trajectories from Implicit Dynamics. Figure 9 are latent trajectories from Brownian motion. Figure 10 are latent trajectories from Variational Auto-Encoder.

## J WHY DOES VARIATIONAL AUTO-ENCODER PERFORM WELL ON THE RECIPE DOMAIN?

In Table 5, VAE does much better than TC. Here we investigate potential reasons why.

```
[ USER ] I'm interested in buying some tickets for a movie tonight. [
ASSISTANT ] OK. Can you tell me the city and state where the theater is
located? [ USER ] It's been so long since we've been to the movies that I want
to get out. It can't get any better. [ ASSISTANT ] Great. Can you tell me the
city and state where you plan to see the movie? [ USER ] I'll be in Santa
Barbara, California at the moment. [ ASSISTANT ] Got it. Are you interested in
a particular movie? [ USER ] There's a couple of nice theaters in town. [
ASSISTANT ] OK. Is there a movie you have in mind? [ USER ] I'm thinking
action. [ ASSISTANT ] Ok. Let's see what action films are out in that area. [
ASSISTANT ] I found one action movie at AMC Mountain 16: No Time To Die. Are
you interested in that one? [ USER ] My best friend Sally has been wanting to
see that one, and I guess we'll go with that one. [ ASSISTANT ] OK. Are you
interested in this movie though? [ USER ] Definitely interested in seeing No
Time to Die. [ ASSISTANT ] OK. Remaining showtimes for No Time To Die at AMC
Mountain 16 are 4:30pm, 6:40pm and 9:10pm. [ USER ] 9:10 pm is good. [
ASSISTANT ] Got it. And how many tickets? [ USER ] Just me and a couple other
guys. [ ASSISTANT ] All right. Let me confirm that you'd like three tickets
for No Time To Die at AMC Mountain 16 tonight at 9:10pm. Is that all correct?
[ USER ] Yes that's right. [ ASSISTANT ] Is it OK to go ahead and purchase
these tickets? [ USER ] Yes go right ahead. [ ASSISTANT ] OK. Your tickets are
purchased and details for how to proceed have been sent to your email address.
Can I help with anything else? [ USER ] No. [ ASSISTANT ] OK. Enjoy your
movie! As always, have a good day. javascript ive been! havn bout ive heard
it. dont think to much. in there. lmao. id appreciate it. reccommended. The
email address for your tickets is theater. There's no problem at all with
anything else. thanks! sign off. Your tickets are confirmed and they've been
sent to your email address. thanks a lot. your welcome. enjoy your movie.
-lbye. I'm very, very sorry for leaving you behind. Thanks for calling. Is
there anything else I can help with? dont remember the last time we went out.
I'm very sorry. Thank you very much. arn. Thanks for all of your help. Enjoy
the movie. wwww, bye. -en Chinese girlfriend, bye. can you enjoy your movie.
Bye. remaining showtimes for No Time To Die at AMC Mountain 16 are 4:30pm,
6:40pm and 9:10pm. This is all for right now. N/A Thanks for all of your
assistance. @ @ b/r/r/r/r abys mam. @ cornyyyyyyyyyyy. BULLY BULLY MOVIE FOR
YOU! Thank you for calling. Enjoy your movie! Your movie. Aww, bye. happy
movie. thanks for calling. -en . Thanks for using our services. > .
```

Figure 4: Example 2 of GPT2 forced long text generation.

We looked into comparing the latent structure recovered by the VAE baseline and TC; see Figure 11.
What we found is the Time Control is best at extracting time over the course of the document (ie.
its latents recover a Bridge process correlated with time), whereas the VAE doesn't elicit strong
temporal structure. We hypothesize that temporal structure is not all you need to succeed in the
Recipe domain.

## K  ADDITIONAL EXPERIMENTS

On discourse coherence, we ran ALBERT on $k = 1$ since ALBERT is trained on this setting. We
found that it got slightly above random chance, unlike the most of the baselines we compare against.
The discourse accuracy is $50.1 \pm 9.0$ on Wikisection, $55.8 \pm 4.0$ on TicketTalk and $60.8 \pm 2.8$ on
TM-2.

```
[ USER ] Hey! I wanna go tonight to watch a movie! [ ASSISTANT ] Hi, what movie would
you like to watch? [ USER ] I want to know about Birds of Prey! [ ASSISTANT ] Sure, is
there any particular theater you would like to watch Birds of Prey at? [ USER ] Can
you tell me what theaters are showing the movie? [ ASSISTANT ] What are you looking
at? [ USER ] Can you tell me about what theaters are showing Birds of Prey at? [
ASSISTANT ] Are you interested in one theater? [ USER ] What theaters are showing
Birds of Prey at? [ ASSISTANT ] Okay, what movie theatre are you going to? [ USER ] I
want to go to AMC. [ ASSISTANT ] Very well, what time would you like to watch the
movie? [ USER ] What is the movie playing tonight at 1:00pm and 3:00 pm? [ ASSISTANT ]
Can you look up times for Birds of Prey at AMC theaters? [ USER ] Can you tell me
about the time and the theater where Birds of Prey is playing? [ ASSISTANT ] Sure,
would you like to watch an IMAX, 3D or Standard showing? [ USER ] I want the IMAX show
especially if there is an IMAX showing. [ ASSISTANT ] Sure, how many tickets would you
like to purchase? [ USER ] I need five tickets. [ ASSISTANT ] Okay, I found the IMAX
showing at 1:00pm, does that work for you? [ USER ] Sure, let's do that! [ ASSISTANT ]
How many tickets would you like to purchase? [ USER ] I want two tickets, for me, my
daughter and my doctor. [ ASSISTANT ] Okay, I have purchased two tickets for Birds of
Prey at AMC theater for 1:00pm for three people. Is that all correct? [ USER ] Yes,
that's correct. [ ASSISTANT ] Okay, would you like to confirm your purchase and book
your tickets? [ USER ] Yes, I want to purchase tickets [ ASSISTANT ] Please provide me
your email address, I'll see what I can find. [ USER ] I'd like to use google, is
there an information available for emailing tickets to birds of prey? [ ASSISTANT ]
Sure, would you like me to check where my tickets are located? [ USER ] Can you tell
me the closest theaters to me for tickets showing tonight? [ ASSISTANT ] Sure, how
would you like to see the movie playing in theaters near you? [ USER ] Can you look
for any similar theater, get me food, and tickets? [ ASSISTANT ] Okay, showing at AMC
theaters in Davis, California, is there a way you can search? [ USER ] Look up some
movie showtimes for me, see if they have anything that is showing in IMAX or 3D. [
ASSISTANT ] Would you like me to search all theaters for tickets showing any available
IMAX shows? [ USER ] Can you look for showing times of Dolittle at AMC? [ ASSISTANT ]
Sure, could I look at the showtimes for the movies showing at 3D? [ USER ] I want the
earliest showing times, maybe at 10:20 pm. [ ASSISTANT ] Not a problem, not an issue,
let me know if I can change that. Thanks, I'll get it on my phone. C'mon, have you
decided on a time? [ ASSISTANT ] Yes, I saw your email for confirmation and booked
three tickets for Birds of Prey at AMC theater for 1:00 pm, does that work for you? [
USER ] Oh ok, forget about it now. [ ASSISTANT ] Would you like to checkout the
booking details and book your tickets online, please. [ USER ] Actually, can you help
me find tickets for Dolittle showing at 10:20 pm in 3D instead of 3D? [ ASSISTANT ]
Not a problem, can you look up tickets to that showing instead instead? [ USER ] Can
you think of anything else? [ ASSISTANT ] Sure, I'll check out the tickets at AMC
showing time showing in 3D as well, if there is anything else? [ USER ] Yea, cool, I
just need to get those booked and I'll get more information to help buy those tickets!
[ ASSISTANT ] Awesome, I have just booked three tickets for Dolittle showing in 3D at
AMC showing time in 3D at 10:20 pm tonight, does that work for you today? [ USER ] I
want to upgrade to 3D, and then use my Apple Pay. [ ASSISTANT ] Okay, I'm turning this
over and getting those tickets online now. Back to you, I want to purchase your movie
tickets to purchase the movie tickets tonight at 10:20 pm at AMC showing in 3D.
Correct? [ USER ] Sure, so I want to purchase a movie ticket for the movie showing in
3D instead of 3D instead, and then use my Apple Pay. [ ASSISTANT ] Okay, I will now
see if you can purchase the movie tickets to purchase your movie tickets to purchase
the movie tickets tonight at 10:20 pm at AMC showing in 3D. Is there anything else you
need help with? [ USER ] Not really, not really. Thanks!
```

Figure 5: Example 1 of Time Control forced long text generation.

## L  PREVIOUS EXPLORATORY THREADS

### L.1  SUCCESS AND FAILURE MODES IN FINE-TUNING GPT2

We found that fine-tuned GPT2 was able to replicate certain aspects of a document corpus, such
as section header ordering ( 92% accurate). However, when going from document-level to section-
level statistics, we noticed that GPT-2 seemed to be either undershooting or overshooting the section
lengths. These results are reported in Table 3.

```
[ USER ] Hello. I am thinking of going to see a movie today. [ ASSISTANT ] OK. And
where will you be seeing the movie? [ USER ] Maybe in Creek's End, Oregon? [ ASSISTANT
] Creek's End, Oregon. Got it. Is there a particular movie you have in mind? [ USER ]
Just any James Bond movie. [ ASSISTANT ] Ok. I think you are talking about No Time To
Die. Does that sound right? [ USER ] Yes it is, No Time To Die is the name. [
ASSISTANT ] I think action movies are always a great choice. In Creek's End, Oregon, I
see that No Time To Die is showing in two theaters: AMC Mountain 16 and AMC Mercado
24. [ USER ] I want the AMC Mercado 24 its what I used to call it. [ ASSISTANT ] It
looks like tonight's remaining showtimes for No Time To Die at AMC Mercado 24 are
5:00pm, 7:10pm, and 9:45pm. Which is best for you? [ USER ] Can I change the theater
from the one I picked because the another one has other showings around it? [
ASSISTANT ] OK, No Time To Die it is. [ USER ] Can I have showtimes for two instead of
one? [ ASSISTANT ] Sure, you chose AMC Mercado 24. [ USER ] Can I change the time, I
want the earliest showtime instead? [ ASSISTANT ] Let me check showtimes at AMC
Mercado 24. [ USER ] Can I actually change the showtime to the showtime you said? [
ASSISTANT ] Sure, you chose the showtime at 3:00pm. [ USER ] Okay that works better,
can I change the time again. [ ASSISTANT ] Remaining showtimes for No Time To Die at
AMC Mercado 24 are 4:30pm, 6:40pm and 9:45pm. [ USER ] Can I purchase two tickets to
the showing of No Time To Die at AMC Mercado 24? [ ASSISTANT ] Sure, you would like
two tickets for No Time To Die at AMC Mercado 24 at 3:00pm. Correct? [ USER ] Yes,
please. [ ASSISTANT ] Okay. So to confirm you would like two tickets for No Time To
Die at AMC Mercado 24 at 3:00pm tonight at the AMC Mercado 24 theater? [ USER ] That's
right. [ ASSISTANT ] Is it OK to go ahead and purchase these tickets? [ USER ] Sure,
let's do it. [ ASSISTANT ] OK. Your tickets are purchased and details for how to
proceed have been sent to your email address. Can I help with anything else? [ USER ]
Actually, let me think before I purchase the tickets since it will be a rough night at
work. [ ASSISTANT ] I understand. Could you please detail how I got these tickets? [
USER ] Can I just send them to my email? [ ASSISTANT ] Sure. Let me see what I can
find. [ ASSISTANT ] Unfortunately I cannot find any movie title that matches your
description. [ USER ] Can I get the dealer's list of movie names from the AMC Mercado
24 website? [ ASSISTANT ] Showtimes for No Time To Die at AMC Mercado 24 are 4:30pm,
7:10pm, and 9:45pm. [ USER ] Ok, let me see what I can find. [ ASSISTANT ]
Unfortunately I cannot find any movie title that matches your description. [ USER ] No
I think we have to choose between No Time To Die and King Johnny please. [ ASSISTANT ]
Got it. Let me see what I can find. [ ASSISTANT ] Unfortunately I cannot find any
movie title that matches your description. [ USER ] Ok, well, I don't want to buy any
tickets. [ ASSISTANT ] Showtimes for No Time To Die at AMC Mercado 24 are 3:00pm,
5:10pm, 7:30pm and 9:45pm. [ USER ] Can you fix something else? [ ASSISTANT ] That's
it for now. [ USER ] Thanks. Could you actually order the tickets? [ ASSISTANT ] No
need, I think we are done with this. [ USER ] Can you just order these? [ ASSISTANT ]
Sure. I will be buying these tickets for you at AMC Mercado 24 at 3:00pm today. Is
there anything else you would like to know? [ USER ] Can I make sure my debit card got
charged again? [ ASSISTANT ] Sure. Let me see what I can find. [ ASSISTANT ]
Unfortunately I cannot find any movie title that matches your description. [ USER ]
Ok! No problem. [ ASSISTANT ] Showtimes for No Time To Die at AMC Mercado 24 are
4:30pm, 7:10pm, and 9:45pm. [ USER ] Ok great, I would like to make this change. [
ASSISTANT ] Is there anything I can help with now? [ USER ] Please, could you send the
tickets to my email? [ ASSISTANT ] Sure. I am glad to help. Let me know what you need
help with.
```

Figure 6: Example 2 of Time Control forced long text generation.

L.2  INVESTIGATING THE IMPORTANCE OF DISTANCES BETWEEN SAMPLED SENTENCES

In earlier iterations of the work, we did explore learning with pairwise contrasts with fixed t-distances, e.g. distances of length 1, 5, and 10 sentences. We observed that the resulting latent trajectories elicited different fits to Brownian bridge dynamics, and the quality in fit varied in t across domains; we've include some examples in Figure 12 on Wikisection.

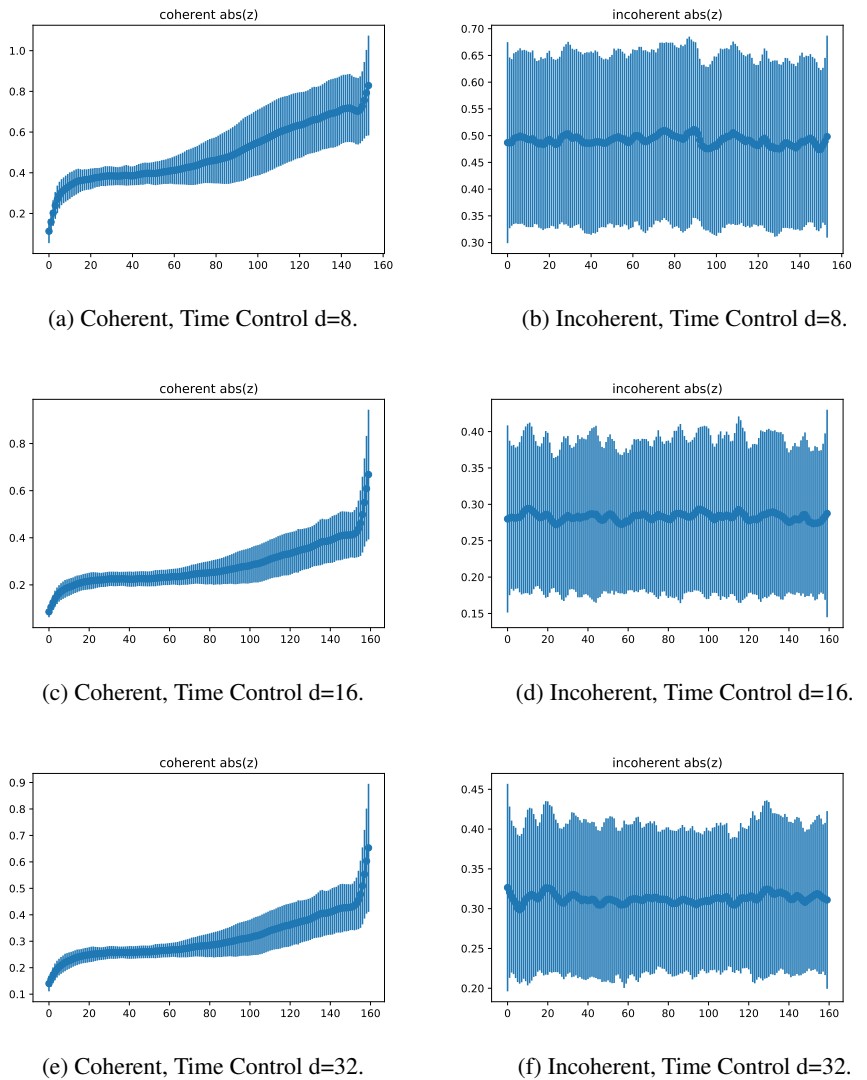

Figure 7: Time Control's latent trajectories over coherent vs. incoherent (randomly scrambled) held-out Wikisection documents. The encoder learns Brownian bridge-like latent trajectories over the coherent documents. The incoherent documents map to noisy trajectories that don't evolve over time.

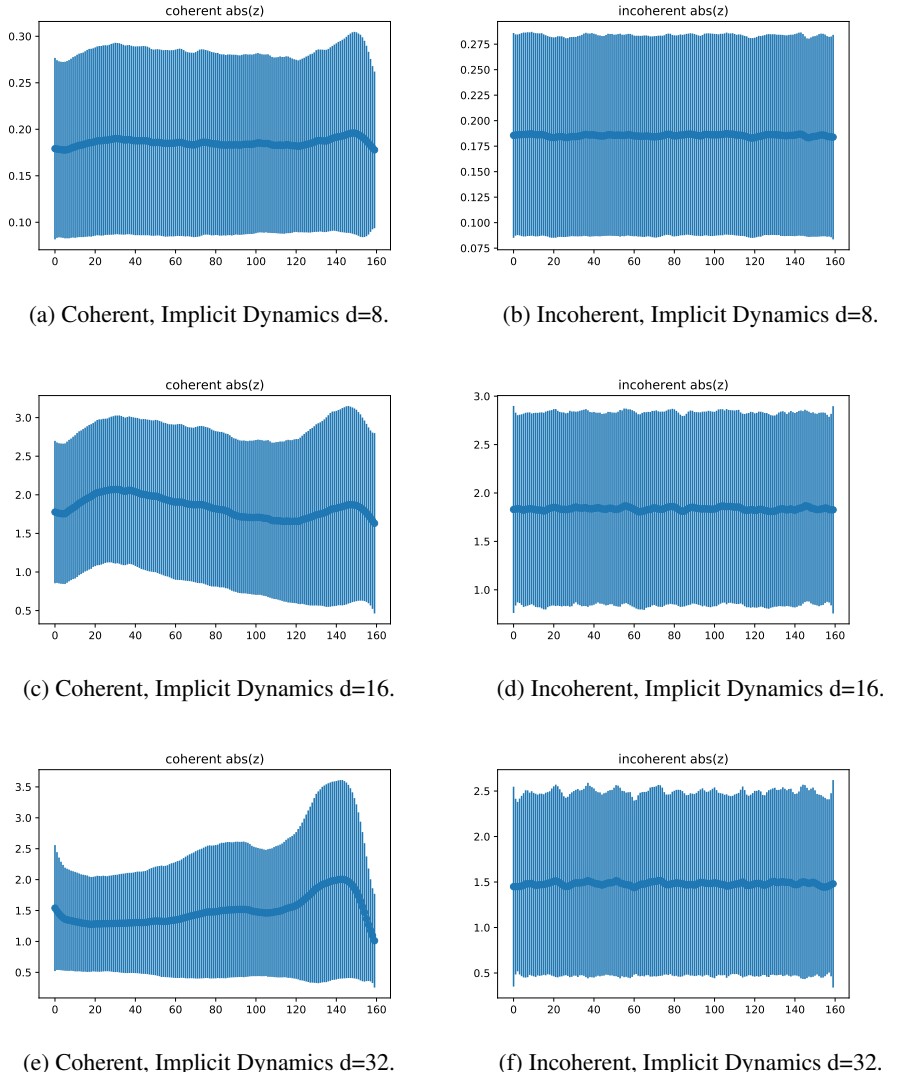

(a) Coherent, Implicit Dynamics d=8.

(b) Incoherent, Implicit Dynamics d=8.

(c) Coherent, Implicit Dynamics d=16.

(d) Incoherent, Implicit Dynamics d=16.

(e) Coherent, Implicit Dynamics d=32.

(f) Incoherent, Implicit Dynamics d=32.

Figure 8: Implicit Dynamics's latent trajectories over coherent vs. incoherent (randomly scrambled) held-out Wikisection documents.

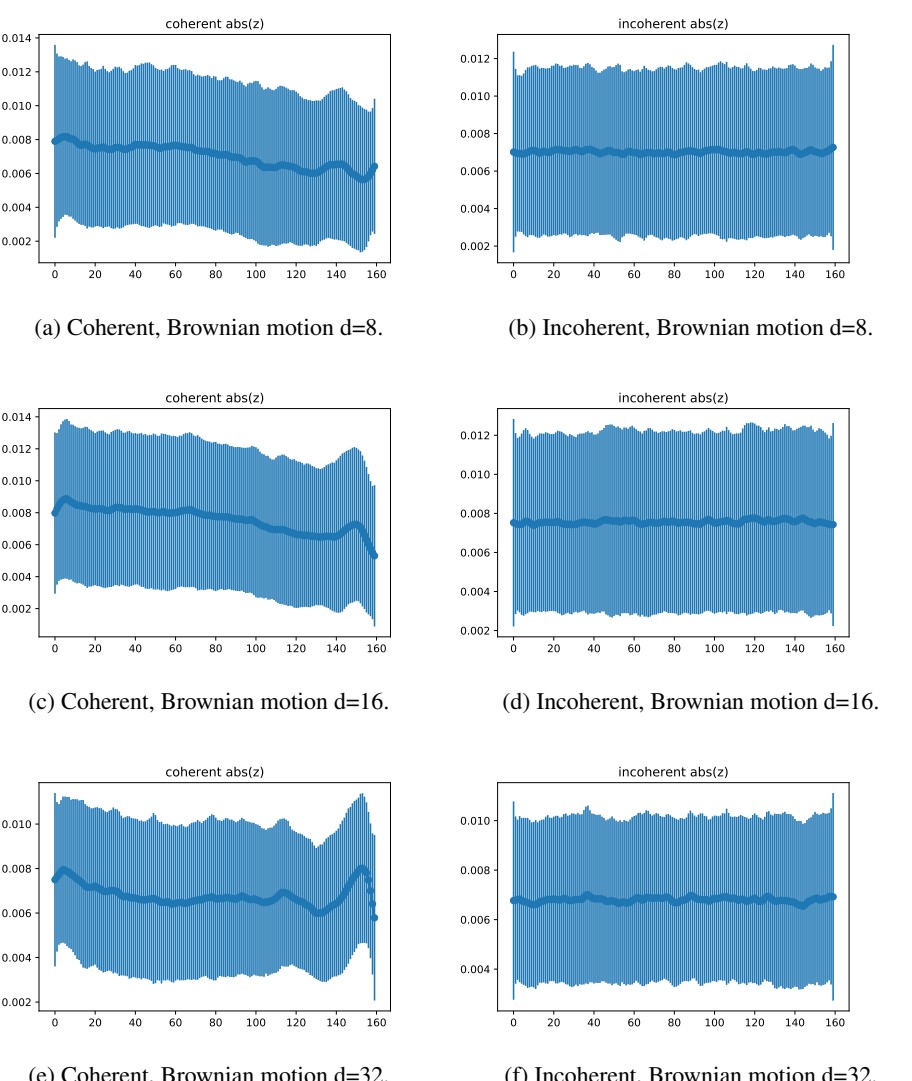

Figure 9: Brownian motion's latent trajectories over coherent vs. incoherent (randomly scrambled) held-out Wikisection documents.

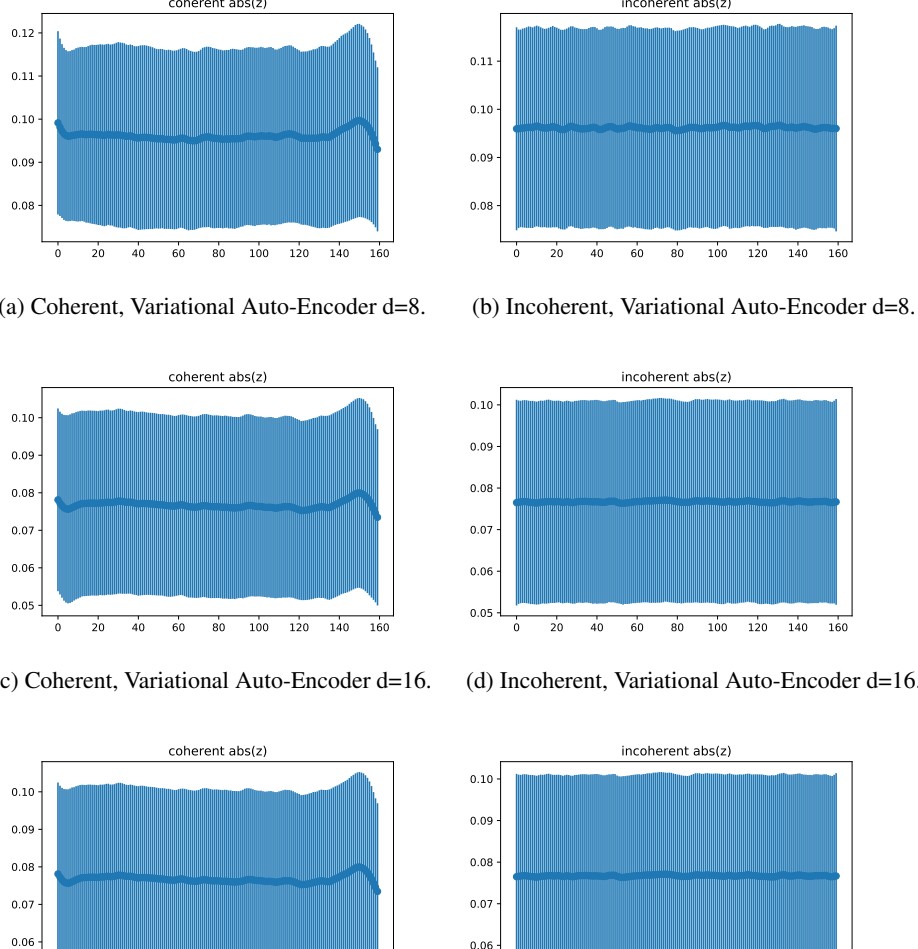

(a) Coherent, Variational Auto-Encoder d=8.  (b) Incoherent, Variational Auto-Encoder d=8.

(c) Coherent, Variational Auto-Encoder d=16.  (d) Incoherent, Variational Auto-Encoder d=16.

(e) Coherent, Variational Auto-Encoder d=32.  (f) Incoherent, Variational Auto-Encoder d=32.

Figure 10: Variational Auto-Encoder's latent trajectories over coherent vs. incoherent (randomly scrambled) held-out Wikisection documents.

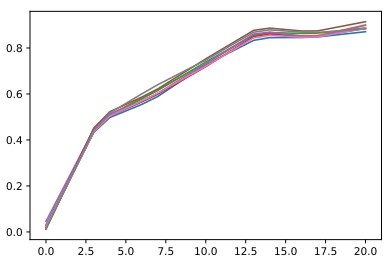 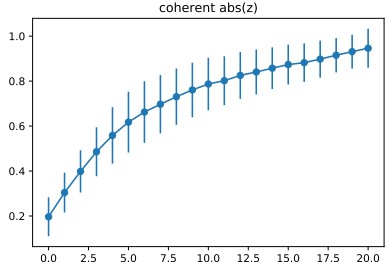

(a) Latent trajectories across dimensions for TC (d=8).

(b) $\mu \pm \sigma$ in latent trajectories across dimensions for TC (d=8).

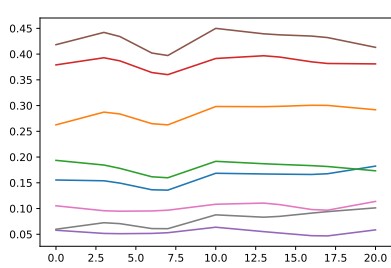 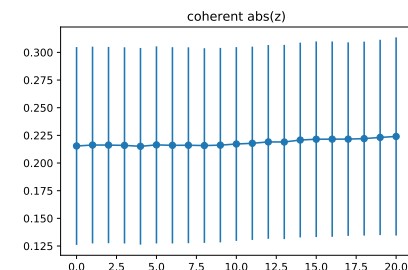

(c) Latent trajectories across dimensions for VAE (d=8).

(d) $\mu \pm \sigma$ in latent trajectories across dimensions for VAE (d=8).

Figure 11: Comparing the latent structure recovered by the VAEbaseline and TC on held-out Recipe documents. Noticeably, TC learns temporally relevant activations whereas VAE latent does not correlate with time.

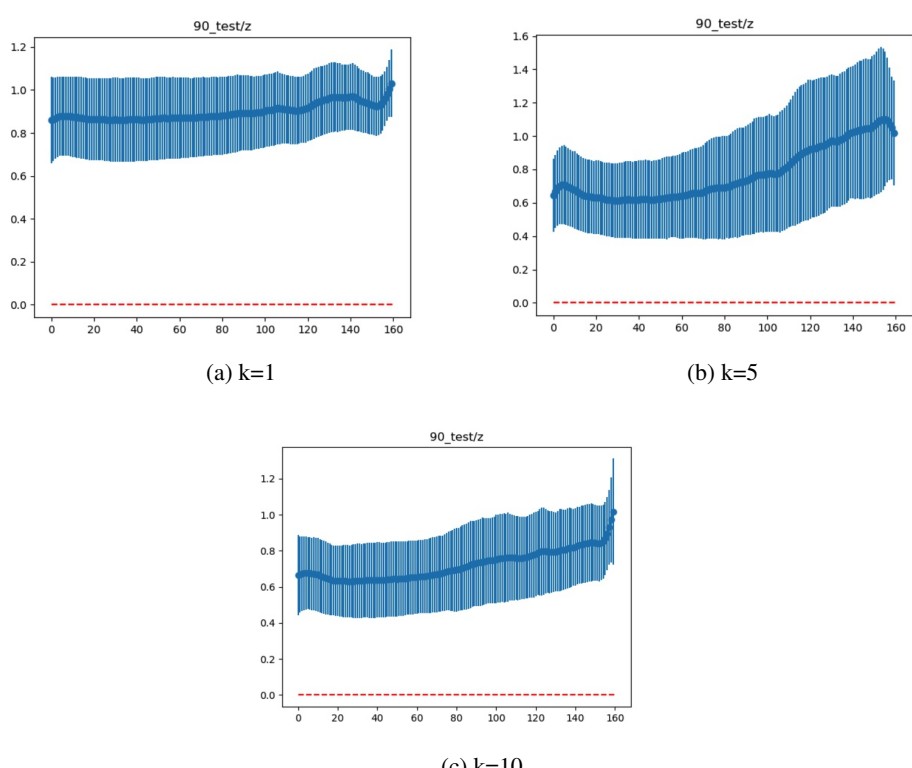

(a) k=1

(b) k=5

(c) k=10

Figure 12: Recovered latent trajectories on held-out Wikisection documents where we used pairwise contrasts with varying time distances; this was on TC, d = 8. Notice how the recovered latent structure varies depending on $k$, the distance between sampled sentences.

| Method | Latent dim $d$ | BERTScore | | | ROUGE | | | BLEURT |
| | | Precision | Recall | F1 | 1-F1 | 2-F1 | L-F1 | |
|---|---|---|---|---|---|---|---|---|
| LM | - | 0.45 | 0.50 | 0.47 | 22.6 | 1.8 | 14.0 | 0.31 |
| ILM | - | 0.50 | 0.51 | 0.50 | 21.6 | 1.8 | 20.7 | 0.33 |
| VAE | 8 | 0.21 | 0.26 | 0.21 | 1.1 | 0.9 | 1.1 | 0.27 |
| VAE | 16 | 0.17 | 0.25 | 0.18 | 1.4 | 0.1 | 1.5 | 0.26 |
| VAE | 32 | 0.10 | 0.10 | 0.12 | 0.2 | 0.6 | 0.2 | 0.26 |
| InfoNCE | 8 | 0.22 | 0.29 | 0.23 | 7.7 | 1.3 | 7.4 | 0.18 |
| InfoNCE | 16 | 0.18 | 0.28 | 0.20 | 1.9 | 0.0 | 1.9 | 0.18 |
| InfoNCE | 32 | 0.20 | 0.28 | 0.21 | 3.0 | 1.0 | 5.8 | 0.18 |
| TC (Ours) | 8 | 0.51 | 0.51 | 0.51 | 17.5 | 1.6 | 16.1 | 0.30 |
| TC (Ours) | 16 | 0.47 | 0.49 | 0.49 | 15.9 | 1.5 | 14.3 | 0.34 |
| TC (Ours) | 32 | 0.50 | 0.50 | 0.50 | 18.4 | 1.4 | 17.1 | 0.32 |

Table 17: BERTScore (Zhang et al., 2019), ROUGE, and BLEURT (Sellam et al., 2020) on ground truth infilled sentence and the generated sentence.

