# OpenReview forum: "Language modeling via stochastic processes"
_ICLR.cc/2022/Conference — ICLR 2022 Oral_

### Official Review · Reviewer_HWGg · 2021-10-26

**Correctness:** 3
**Technical Novelty And Significance:** 3
**Empirical Novelty And Significance:** 3
**Recommendation:** 8
**Confidence:** 4

**Main Review:**

I really like the main modelling contribution of this paper. It is this reviewer's personal opinion that to do long-form text generation, it is not enough to generate token-by-token, but that some high-level planning is required, and the Brownian bridge process model (Time Control; TC) the authors propose is definitely a good candidate to model the latent drift of discourse (indeed, papers like [1] already used random walk-style models to explain properties of word vectors). There are some prior works on using structured probabilistic models, such as switching latent dynamical systems, for text generation [2], which should also be cited.

The motivation of the model present is clear, and the description of how the model is trained is generally clear enough to reimplement. It wasn't immediately clear that training the model on triples only is enough to guarantee general Brownian bridge dynamics for the entire text trajectory, I feel a note should be added to clarify this. My other quibble here is with how the model is presented: although the general probabilistic model is written down in Equation 1, the likelihood function (i.e. the functional form of p(z_t | x_0, x_t, x_T)) is not explicitly written down anywhere, which leads to confusing things like the variance of the process \sigma^2 being used in Equation 3 without prior introduction. I feel like explicitly writing down the likelihood would make the equations in the paper flow much better.

I feel the major weakness of this paper is with the experimental sections. For various reasons, I have objections to each of the experiments, which I will go through below:

- The first experiment attempts to show that TC is a better model of local discourse coherence. The authors take two sentences from a document k steps apart, embeds them and them attempts to predict the sentence ordering from the embeddings. They say that for k=1 all models considered perform at chance level on all datasets, and only show results for k=5 and k=10. However, models trained using the k=1 objective (such as ALBERT [3] and StructBERT[4]) seem to be able to perform the task better than chance, so theoretically this should be possible. Therefore, I think the baselines should at least include an ALBERT model to show the performance upper bound on this problem. Further, k=5 (or even 10) starts meaning the sentences start becoming very far apart (10 dialogue turns is more than enough to complete some of the simple dialogue agent tasks!), so it's questionable whether the model is really modelling 'local' dynamics at this point.

- The second experiment looks at text infilling on the ROCStories dataset, and use BLEU and BLEURT to automatically evaluate their models (although the BLEURT results do not appear to be anywhere in the paper). The reported BLEU results are really low, to the extent that it's unclear whether an improvement from 2 to 5 BLEU is really meaningful. Part of the issue is that BLEU measures precision, which penalises text generation where there are a variety of possible outputs; for this reason, [2] report ROUGE results on ROCStories, which are much better. The missing BLEURT results would help contextualise model performance here. The human evaluation shows the model performs about as well as the ILM baseline from [5], which is ok I guess?

  In addition, the table ordering is incredibly confusing. Table 6, which shows the human evaluation for experiment 2, appears much later in the text, after tables for the later experiments. It took me a long time to find it. Can you group the tables a bit better, in thematic order?

- The third experiment attempts to measure 'global text dynamics' by measuring length mismatch per section on Wikisections. It's unclear what notion of 'global text dynamics' the authors are referring to - there are many theories on discourse coherence of long text, and none of them easily map onto a simple measure of section length. If the authors simply mean whether the model has learnt a notion of document structure, I think it would be better to be more explicit about this: showing that fine-tuned GPT2 can't even replicate the structure of a homogenous document corpus is an interesting negative result.

- The fourth experiment forces models to generate beyond the expected document length by suppressing generation of the EOD token. I'm really not a fan of this experiment, because I don't even expect TC to perform well on it. Do the authors just keep on conditioning the decoder on z_T, and force the model to generate from this? At this point, the model is just a standard autoregressive model, so the modelling contribution should have no effect. Alternatively, do the authors resample z_{t+1} each time the model finishes generating a sentence? In which case, how do the authors preserve the Brownian bridge dynamics, conditioning on hitting a target state z_T? There are a few methodological issues with this experiment. A better experiment to run would be to simply ask the human annotators to score texts freely generated from GPT2 and TC for coherence, as a measure of how well TC can generate coherent text.

Overall, while the experimental section is weak, I really believe the core idea of directed Brownian dynamics for planning is a cool one, and deserves to be shared more widely. This is why I recommend acceptance.

References:
- [1]: RAND-WALK: A latent variable model approach to word embeddings, Sanjeev Arora et al. 2015
- [2]: Generating Narrative Text in a Switching Dynamical System, Noah Weber et al. 2020
- [3]: ALBERT: A Lite BERT for Self-supervised Learning of Language Representations, Zhenzhong Lan et al. 2021
- [4]: StructBERT: Incorporating Language Structures into Pre-training for Deep Language Understanding, Wei Wang et al. 2021
- [5]: Enabling Language Models to Fill in the Blanks, Chris Donahue et al. 2020


==================

Post author response:

> Nonetheless, we think these observations fit well with the intuition our work proposes: Neighboring sentences are close to each other and act like Brownian motion where ordering is difficult to infer, and goal-orientedness / discourse structure emerges on longer stretches of sentences in a document.

I like this framing - currently it's implicit in the paper, but maybe it can be made more explicit that we expect the larger k results to be better, and that this verifies the Brownian bridge approach towards modelling text dynamics.

> Nonetheless, the end arbiter of this task is a human (how coherent do the generations sound to a human?) and we care about at least matching ILM, a method developed specifically for text-infilling. So, it’s promising that our method performs better and/or competitively with ILM on human-based metrics (BLEURT and Human evaluations in Table 6).

I think it should be made explicit then that ILM is in effect an upper bound for model performance, as it is a model trained specifically to do the task, and that matching the performance of ILM is actually a strong result for the TC model.

> So, to directly answer the reviewer’s question: we do not condition the decoder on z_T and do not resample new latents during generation.

The model is thus primed to generate much longer text than it was typically exposed to? Thank you for the clarification.

> We in fact do already ask human annotators to score the generation (rf. Table 7). In this setup, we remove the middle section of the generated output as the text is extremely long. See Figures 3-6 for examples of the full forced long text generation results.

I believe the stronger (and more realistic) human evaluation is to not just evaluate the tail coherence on forced long text generation, but  instead directly sample from the model naturalistically and evaluate that output using human annotators. If TC better captures global coherence, this should be visible even in this setting.

Overall, I would like to thank the authors for their response. Many of my concerns have been addressed, and I am happy to increase my score.

**Summary Of The Paper:**

The authors propose to use a Brownian bridge process to model global coherence of a long piece of text. They show how to train such a model in an encoder-decoder style setup, using a contrastive loss to model the Brownian bridge dynamics. The authors then verify aspects of their model with a series of experiments to show that their model with an underlying generative process outperforms competing approaches on a variety of local and global coherence and generation tasks.

**Summary Of The Review:**

Interesting modelling contribution to ensure global coherence of generated text. The proposed modelling approach could have wide applicability, which is why I recommend acceptance.

---

> ### Author Response · Authors · 2021-11-16
> **Thank you! We've revised the paper to include additional experiments (ALBERT, ROUGE, BLEURT), discussions on BB dynamics, and clarification on decoding.**
>
> We thank the reviewer for their feedback and questions! We’ve revised the paper to include results from additional experiments suggested by the reviewer (ALBERT baseline and ROUGE/BLEURT metrics), discussions on guaranteeing Brownian bridge latent dynamics, elaboration on the global text dynamics setting, and decoding setup in the forced long text generation setting.
>
> > There are some prior works on using structured probabilistic models ... which should also be cited.
>
> Thank you for the suggested citations! We were not aware of these works. We’ve edited the related works section to reflect these updates.
>
> > It wasn't immediately clear that training the model on triples only is enough to guarantee general Brownian bridge dynamics for the entire text trajectory, I feel a note should be added to clarify this.
>
> We agree with the reviewer that training on triplet samples may not guarantee Brownian bridge dynamics for the entire text trajectory.
> On the empirical side, we found it easier to recover Brownian bridge dynamics via contrast triplets than the ELBO objective or with pairwise contrasts (rf. Appendix J and M.2). We’ve included a note on this in the paper revision.
>
> > My other quibble here is with how the model is presented … which leads to confusing things like the variance of the process \sigma^2 being used in Equation 3 without prior introduction.
>
> We thank the reviewer for their suggestions on presenting the model. We’ve updated the text to clarify that the variance of the process is the variance presented in Equation 1: we scale the distance in Equation 3 by the relative sentence location. Hopefully this clarifies the inference process for z_t.
>
> > I think the baselines should at least include an ALBERT model to show the performance upper bound on this problem.
>
> Thank you for this suggestion! We reran the discourse coherence experiments for k=1, 5, 10 with ALBERT on the three domains presented in Table 1 (Wikisection, TM-2, TicketTalk).
>
> As the reviewer hypothesized, we found that on k =1, ALBERT indeed performed above random chance on the conversation-based domains (55-60%) however performed randomly across all k-values on Wikisection. We believe this is due to the lack of discourse signal in fact-oriented domains like Wikipedia articles. We’ve included the k=1 results in Appendix L and updated Table 1 to include the k=5 and 10 results.
>
> > [I]t's questionable whether the model is really modelling 'local' dynamics at [k=5 or 10 in dialog settings].
>
> We agree with the reviewer that k=5 and 10 results in a much simpler discourse prediction setting for dialog domains. Local dynamics defined with k=5 or 10 varies in discourse implications across domains.
>
> Even so, we found that in the k=1 prediction setting, most (if not all) of the baselines failed to achieve better than random chance. In investigating why this is the case for the dialog domains, we noticed that there were several sentences which contained simple, ambiguous responses, such as “OK” where sentence ordering is difficult to infer from. We hypothesize this is the reason why doing better than random is difficult on k=1. ALBERT is the only baseline that achieves slightly better than random performance, but it’s not significantly better (~55-60% test accuracy).
>
> Nonetheless, we think these observations fit well with the intuition our work proposes: Neighboring sentences are close to each other and act like Brownian motion where ordering is difficult to infer, and goal-orientedness / discourse structure emerges on longer stretches of sentences in a document.

---

> > ### Author Response · Authors · 2021-11-16
> > **Continuation of response**
> >
> > >  the BLEURT results do not appear to be anywhere in the paper ... The reported BLEU results are really low ... [2] report ROUGE results on ROCStories, which are much better. The missing BLEURT results would help contextualise model performance here.
> >
> > Thanks for the suggestion and the reference! We agree with the reviewer’s intuitions with the low BLEU scores: the scores are low due to the open-endedness of the domain. We reran our text infilling experiments for all the methods and tracked BLEURT and ROUGE, as the reviewer suggested. The results are reported in Table 17.
> >
> > As a clarification, our original work did not report BLEURT scores rather BertScore; those results were put in the Appendix (Table 17) due to space constraints.
> >
> > We found that Time Control performs slightly better on BLEURT than the alternatives including ILM, a method specific for text infilling.
> > It performs slightly worse than ILM and LM as ranked by ROUGE (1, 2, L) scores, but does slightly better on the BLEU scores (Table 2). The ROUGE and BLEU results indicate a precision-recall tradeoff where TC generates infill-sentences with more overlap to the ROCStories reference, and the reference words appear more frequently in ILM/LM generation.
> >
> > Nonetheless, the end arbiter of this task is a human (how coherent do the generations sound to a human?) and we care about at least matching ILM, a method developed specifically for text-infilling. So, it’s promising that our method performs better and/or competitively with ILM on human-based metrics (BLEURT and Human evaluations in Table 6).
> >
> > > “The human evaluation shows the model performs about as well as the ILM baseline from [5], which is ok I guess?”
> >
> > Yes, our method is not specific to the task of text-infilling so it’s promising that it performs close to ILM, a method designed for the text infilling task!
> >
> > > “Can you group the tables a bit better, in thematic order?”
> >
> > Thanks for letting us know. We’ve re-organized the tables and the new results suggested by reviewers in our revisions -- please let us know if the updated version improves readability!
> >
> > > “If the authors simply mean whether the model has learnt a notion of document structure, I think it would be better to be more explicit about this”
> >
> > Thanks for pointing this out! We’ve updated the paper to be more explicit about the success and failure modes we ran into over the course of this work in Appendix M.
> >
> > Regarding the concern of ‘modelling document structure,’ we found that fine-tuned GPT2 was able to replicate certain aspects of a document corpus, such as section header ordering (~92% accurate). However, when going from document-level to section-level statistics, we noticed that GPT-2 seemed to be either undershooting or overshooting the section lengths; these are the results that were reported in the original submission.
> >
> > We believe this is an interesting direction for future research to explore: Is the section length mismatch caused by a mismatch between the myopic next-token prediction LM objective and retaining higher-level statistics like section lengths? We’ll be more precise in our revisions about what we mean by global text dynamics.
> >
> > > “Do the authors just keep on conditioning the decoder on z_T, and force the model to generate from this? [...] Alternatively, do the authors resample z_{t+1} each time the model finishes generating a sentence?”
> >
> > Thanks for raising this to our attention! We agree the paper should have been more clear on how we obtain latent plans in the forced long text generation setting.
> > The latent planning process is the following.
> >
> > Let’s have $S_{\text{avg}(\mathcal{D})}$ denote the average number of sentences in a document and $T_{\text{avg}(\mathcal{D})}$ denote the average number of tokens in a document.
> >
> > Rather than planning a trajectory of length $S_{\text{avg}(\mathcal{D})}$ (the average number of sentences in a document) which is what is done in Section 4.3 for normal text generation, we scale the trajectory length to $c \cdot S_{\text{avg}(\mathcal{D})}$. $c$ is determined by how many more tokens we need in order to fill up to GPT-2 maximum context length of 1024: $c = \frac{1024 - T_{\text{avg}(\mathcal{D})}}{T_{\text{avg}(\mathcal{D})}}$.
> >
> > So, to directly answer the reviewer’s question: we do not condition the decoder on z_T and do not resample new latents during generation.
> >
> > We’ve updated the paper to include these explanations.
> >
> > > “A better experiment to run would be to simply ask the human annotators to score texts freely generated from GPT2 and TC for coherence, as a measure of how well TC can generate coherent text.”
> >
> > We in fact do already ask human annotators to score the generation (rf. Table 7). In this setup, we remove the middle section of the generated output as the text is extremely long. See Figures 3-6 for examples of the full forced long text generation results.

---

### Official Review · Reviewer_JgDa · 2021-11-03

**Correctness:** 3
**Technical Novelty And Significance:** 3
**Empirical Novelty And Significance:** 3
**Recommendation:** 8
**Confidence:** 4

**Main Review:**

This paper has an interesting approach and tackles an important problem of streamlining sequence generation from autoregressive models.

The experiments show the value of learning a manifold over the latents that have a linear relationship with some stochastic perturbation. They provide evidence that learning in such a manner is promising in order to maintain coherence over long text generation.

However, the setting is fairly limited because this approach requires two contextual endpoints, the start and finish. This is especially underwhelming given that the introduction states that this approach aims to perform \emph{controllable goal-oriented} generation. In my view, the setting described and experimented with doesn't reflect this goal. For example, there are limited experiments with regard to controllable generation, or goal-oriented generation tasks.

Secondly, the assumption that autoregressive generation follows a Brownian motion is strong and I would like to see some empirical evidence or theoretical argument supporting this. One simple experiment could be to actually try to fit a Brownian motion model to a bunch of sequences generated from GPT-2, and show that this fitted model is not suitable for naturally occurring text.

Experiment wise, my biggest concern is the VAE baseline. The point of this baseline is to show that for the same setup of Brownian bridge process, contrastive learning is better than the VAE objective, but I feel that the VAE implementation as described in the appendix does not make the comparison fair. Due to lack of details in the paper, I am assuming that the priors p(z0) and p(zT) are standard gaussians. If this is not true, then a clarification would ease this concern of mine. But assuming this is true, the loss basically tries to match the encoder distributions q(z0) and q(zT) obtained by f_{\theta}(x0) and f_{\theta}(xT) to the standard Gaussian. What this means is that there is a pressure to make the 0 and T embeddings similar which is not at all what we want from this bridge process model. A more careful instantiation of prior for VAE, or even learning a time sensitive prior would be a better implementation of the VAE baseline.

Table 1 is another concern. This experiment basically trains a linear classifier over the encodings to identify if they are in-or-out of order. The proposed approach is naturally suited for this metric/classifier because the encodings at different times are more or less linearly related with some stochasticity. However, this is not true for the other baselines, so I am not sure what is the takeaway message from this experiment.

Also, more exposition on the Brownian motion baseline would be helpful. The current description is not enough to get an idea about what exactly was done for generation and other experiments with this baseline. On a related point, I don't get why BM for Table 2 would be the same as the brownian bridge. Isn't it the case that Brownian motion baseline doesn't get to see \emph{both} the endpoints? If I am mistaken about this, then more exposition is required here because I checked both the paper and the appendix carefully for this.

Table 5 shows mixed results. More discussion and analysis here would be helpful.

For clarification: please make explicit whether the triplets have a notion of distance or not i.e. it is sensitive to different value of t depending on which sentence in the middle was sampled. From the context, I am assuming this is the case but clarification would be helpful. Also, notation in equation 2's denominator is confusing. Are you summing over all the negative x_{t'}?

**Summary Of The Paper:**

This paper proposes to model the evolution of sentences in a document via a stochastic process; specifically a Brownian Bridge process. The paper start off by assuming that the generated sequences by autoregressive models like GPT-2 follow Brownian motion in that they tend to get incoherent and "meander" in the semantic space. This paper aims to reduce this random behavior by pinning the endpoints of the trajectory and model the generation by Brownian bridge process instead. The key intuition behind this process is that given two endpoints z0 and zT, the evolution of z along time t is a Gaussian with mean that is some linear combination of z0 and zT. This paper models text by training an encoder for sentences x that produces the embedding z by training over triplets (x0, xt, xT) where 0<t<T that encourages zt to follow Brownian bridge dynamics and uses contrastive loss with a negatively sampled x't for training.

The approach is tested for local coherence, long range order sensitivity, and generation of long sequences and is compared against ablative and external baselines. The proposed approach does lead to learning of embeddings that are obtainable via linear combination and this leads to improved performance on sensitivity to sentence order in documents and document generation.

**Summary Of The Review:**

Overall, I think this paper is well motivated and proposes a reasonable solution to improve coherence of model generated text. This is supported by ample experiments but I have serious concerns about some of the crucial experiments and baselines that I have detailed in my main review. Also, I think that the paper could be clearer about its contributions and implementation details.
-----
Post rebuttal: thanks to the authors for the detailed response addressing many of my concerns. My biggest concern about the prior in the VAE baseline is somewhat alleviated given that the the authors used different fixed priors for the two settings. While this could be improved by having learnable priors/better priors, I think the current setting makes the experiments reasonably sound. I have raised my score.

---

> ### Author Response · Authors · 2021-11-16
> **Thank you! We've updated the paper to include more details on ablation, goodness of latent model fit, and model sensitivity to sentence distances.**
>
> We thank the reviewer for their feedback and questions! We’ve revised our paper to improve clarity on the ablation implementation, the goodness of fit with BB vs BM latent models, and the method’s sensitivity to the sentence distance sampled for contrasts.
>
> > " the setting is fairly limited because this approach requires two contextual endpoints, the start and finish...For example, there are limited experiments with regard to controllable generation, or goal-oriented generation tasks."
>
> Thanks for raising this point! In earlier iterations of this work, we did test for other aspects of controllable text generation such as varying document lengths using shorter or longer latent plans. We did indeed find that shorter latent plans resulted in shorter text, and longer latent plans resulted in longer text (without suppressing the EOD token), although there wasn’t a 1-to-1 correspondence between the change in plan length and the change in text length across domains. For example, on the Wikisection, we reduced the latent plan length by 5 and 10 times fewer sentences and observed the average document length decreases from 691 tokens to 543 tokens to 542 tokens long (about 20% reduction). When we increase the latent plan length by 2 to 5 times, the average document length increases from 691 tokens to 814 tokens (~ 18% increase) to 842 tokens (~22% increase).
>
> We do think an interesting extension of this work is to do multi-point/goal conditioning: In principle, Time Control can already do this because intermediate trajectories are also Brownian bridges. In this work, we focus on the two-endpoint case.
>
> > “The assumption that autoregressive generation follows a Brownian motion is strong and I would like to see some empirical evidence or theoretical argument supporting this.”
>
> This is an interesting point -- in our original submission, we provide some empirical evidence in Table 9 which reports test PPL after fine tuning GPT2. We find that fine tuning GPT2 on Brownian bridge latent models results in slightly lower PPL on held-out examples than with Brownian motion latent models. This seems to suggest that using Brownian bridge latent models fit better than Brownian motion ones.
>
> > “A more careful instantiation of prior for VAE, or even learning a time sensitive prior would be a better implementation of the VAE baseline.”
>
> Thank you for pointing this out! We agree with the reviewer that the paper should include more details on the VAE implementation and that the VAE prior matters in the bridge process model for our application. We will clarify in the text that the prior for $z_0$ is 0-centered and for $z_T$ 1-centered, just like in our contrastive learning setting. This would lead to the same time-sensitive prior for both settings.
>
> The paper has been updated to address these concerns regarding clarity on the ablation implementation.
>
> > “[Regarding Table 1, t]he proposed approach is naturally suited for this metric/classifier because the encodings at different times are more or less linearly related with some stochasticity. However, this is not true for the other baselines, so I am not sure what is the takeaway message from this experiment.”
>
> The takeaway message from this experiment is to compare against non-temporal baselines: How well of a model fit is the Brownian bridge latent model which assumes temporal dynamics to models that don’t assume temporal dynamics like BERT or SimCSE? Note that our latent model assumptions don’t necessarily have to be true: For example, on the recipe domain where all of the latent models do not perform better than random, these results seem to suggest that neither model assumption is a good way to model text dynamics.
>
> > “Also, more exposition on the Brownian motion baseline would be helpful. [...] On a related point, I don't get why BM for Table 2 would be the same as the brownian bridge.”
> Thanks for raising these concerns. We will update the text to include more exposition to the Brownian motion baseline.
>
> The text infilling setting uses the average of the prefix and suffix sentence embeddings ($\frac{z_{\text{prefix}} + z_{\text{suffix}}}{2}$) for decoding the infill-sentence; one reason we do this is to compare the quality of interpolated latents which is of particular relevance for testing decoding quality between TC and ID (explicit vs implicit dynamics assumption).
>
> In the Brownian Motion case, once we condition the intermediate latent with a target endpoint $ z_{\text{suffix}}$, it becomes a Brownian Bridge by definition. On the other hand, throwing away the endpoint (aka the suffix) in the Brownian motion case felt like an unfair baseline.

---

> > ### Author Response · Authors · 2021-11-16
> > **Continuation of response**
> >
> > > Table 5 shows mixed results. More discussion and analysis here would be helpful.
> >
> > Thanks for raising this concern! We assume by “mixed results” the reviewer is referring to the VAE ablation performing well on the Recipe domain, but less well on other domains.
> >
> > We don’t have full insight as to why the VAE does better on the Recipe domain, but we looked into comparing the latent structure recovered by the VAE baseline and Time Control. What we found is the Time Control is best at extracting time over the course of the document (ie. its latents recover a Bridge process correlated with time), whereas the VAE doesn’t elicit strong temporal structure; see the newly added Appendix K. We hypothesize that temporal structure is not all you need to succeed in the Recipe domain.
> >
> > > “[p]lease make explicit whether the triplets have a notion of distance or not i.e. it is sensitive to different value of t depending on which sentence in the middle was sampled.”
> >
> > We train the encoder to embed sentences such that the distances are scaled w.r.t the ground truth time value with the variance of the Brownian Bridge process, $\frac{t(T-t)}{T}$ (rf. Equation 3). At inference time, the sentence index is not provided: A sentence is directly passed through the encoder without any other supervision. The triplet of sentences are sampled uniformly at random $(x_{t_a}, x_{t_b}, x_{t_c})$, then re-ordered such that $t_a < t_b < t_c$; this means that the variance does change across triplet samples.
> >
> > In earlier iterations of the work, we did explore learning with pairwise contrasts with fixed t-distances, e.g. distances of length 1, 5, and 10 sentences. We observed that the resulting latent trajectories elicited different fits to Brownian bridge dynamics, and the quality in fit varied in t across domains; we’ve included some examples in Appendix M of the revised paper in case this is of future interest.
> >
> > These observations seem to complement the discourse coherence results and how the difficulty of inferring discourse structure varies on the distance between sentences and the domain (eg. larger distances are easier in dialog settings).
> >
> > We've updated the text to make this more clear.
> >
> > > “Are you summing over all the negative x_{t'}?”
> >
> > Yes, we are summing over all negative $x_{t'}$. We’ve updated the text to make this more clear.

---

### Official Review · Reviewer_mT85 · 2021-11-03

**Correctness:** 4
**Technical Novelty And Significance:** 4
**Empirical Novelty And Significance:** 4
**Recommendation:** 8
**Confidence:** 4

**Main Review:**

This paper proposes a generation from a language model not only from an initial
state, but also using a goal state. Instead of Brownian motion, the authors
employ a draw from Brownian bridge by designating initial and end states,
called Time Control.
Experimental results show the proposed generation from Brownian bridge is more
natural and coherent for text-infilling task, and also preserves text
structures both by automatic evaluation and human evaluation.

Using Brownian bridge is a very simple and effective idea for text generation.
My only concern is the range of its applicability: while it is far more natural
than a simple random walk, Time Control only allows designating the first and
last states for generation. However, in the actual situation, it is not always
the case for the first (and sometimes, last) sentence should have a designated
state. First few sentences might constitute just an ice-break, and the actual
content might start after that.
More generally, it is more desirable that we can condition the generation at
arbitrary time. In fact, I think that this can be done by a conditional draw
from a Gaussian process. Since Brownian motion corresponds to using an
exponential kernel of GP, sentence generation from conditional GP would be
the way for the future extention of this work.
Anyway, this work will surely pave the way for such principled generations.

Minor

- Some tables are located within the main text. Tables and Figures should be
placed top or bottom of the paper for readability: please use \begin{figure}[t]
for something like that.
- Numerical results in Tables can be rendered in a smaller font (i.e. \small).
Also I recommend to condense line spacing for Tables for readability, using
\usepackage{setspace} and begin{spacing}{0.9} ... end{spacing}, for example.


**Summary Of The Paper:**

This paper proposes a generation from a language model not only from an initial
state, but also using a goal state. Instead of Brownian motion, the authors
employ a draw from Brownian bridge by designating initial and end states,
called Time Control.
Experimental results show the proposed generation from Brownian bridge is more
natural and coherent for text-infilling task, and also preserves text
structures both by automatic evaluation and human evaluation.

**Summary Of The Review:**

Nice attempt for random generation from neural language models using the idea of Brownian bridge. This work will pave the way for more princpled random generation from language models.

---

> ### Author Response · Authors · 2021-11-16
> **Thank you! We've updated our paper to reflect the reviewer's recommendations.**
>
> We thank the reviewer for their feedback and questions! We’ve revised our paper to include formatting changes recommended by the reviewer.
>
> > Using Brownian bridge is a very simple and effective idea for text generation. My only concern is the range of its applicability: while it is far more natural than a simple random walk, Time Control only allows designating the first and last states for generation. However, in the actual situation, it is not always the case for the first (and sometimes, last) sentence should have a designated state.
>
> Thanks for your feedback and kind words! We concur with the reviewer that an interesting future direction is ‘multi goal-conditioned generation,’ and we intend to investigate as an extension of our work.

---

### Official Review · Reviewer_zz52 · 2021-11-05

**Correctness:** 4
**Technical Novelty And Significance:** 3
**Empirical Novelty And Significance:** 3
**Recommendation:** 8
**Confidence:** 4

**Main Review:**

**Pros**

- The paper is well structured and easy to follow, the idea of modeling sentences to a Brownian bridge latent space is neat and generic enough to (1) allow for noise given its stochasticity (2) doesn't require explicit domain knowledge for planning.

- Well Structured Experiments sections with 4 RQs and results that confirm each of the hypotheses

- Reproducibility and Transparency in reporting of experiments in terms of available source code, dataset information, details about human evaluation, generation examples.

**Areas of Enhancement & Questions to authors**

- The information about each of the ablations (ID, BM) could be explained better. namely the section "ablations".

- There's a clear Inconsistency in the best TC method between different latent dimensions (8,6,32), in most of the experiments there's at least one of the 3 that is performing drastically worse than the other baselines, while there's overall no clear winner. I wonder if you have thoughts about this.

- Table 5 the VAE(32) method performs the best overall in "Wiki section" although the TC (16) method has been highlighted as the best. Is there a reason behind this?

- During the training of the decoder how do you make sure that the decoder uses the information given by the latent plan?

- Overall the paper would have benefited from an intrinsic visualization of the latent space, to make sure for example that there's no  Information collapse of the embeddings when dealing with long sentences. This could be done by visualizing the planning trajectory difference between coherent and incoherent text.




**Summary Of The Paper:**

This paper introduces a method to enhance the global coherence of text generated from Language models. The proposed method (Time Control). Under the assumption in the latent space of sentence embeddings, the incoherent text can be seen as "Brownian motion" in the latent space. In order to enforce a goal to the generated text authors by fixing a start and end to this Brownian motion the process of text generation can be modeled as a Brownian bridge.

From this assumption, the authors drive a method that consists of three steps (1) training an encoder to map sentences to a latent plan defined as Brownian bridge (2) training a decoder to reconstruct sentences from the given context + the true encoded vector of the target sentence from planning latent space using the trained encoder (3) at inference time: given a start and endpoint, a target trajectory of vectors $z_0, ..., z_t, ..., z_T$ is sampled and use the decoder to generate a sentence based on this bridge.

Authors run several experiments to (1) evaluate the hypothesis that the encoder can capture local text dynamics using sentence order prediction task (2) evaluate the decoder to generate local incoherent text using the text-infilling task. (3) capture global text statistics by measuring the statistics (length of Wikipedia sections for city articles) of the generated text and compare them to the ground truth. (4) Evaluate the overall coherence of the long-generated text. Overall the results look convincing except for some caveats (see the areas of enhancement)


**Summary Of The Review:**

The paper introduces a simple method of preserving coherence in language modeling it builds on previous work that tried to implicitly model planning dynamics. The introduced solution is effective and general enough to not need domain-specific planning information. It is a good paper to accept overall. I advise the authors to clarify the information about the used baselines in a more clear manner.

---

> ### Author Response · Authors · 2021-11-18
> **Thank you! We've updated the paper to include additional discussion on the method, visualization of latent trajectories, and fixes suggested by the reviewer.**
>
> We thank the reviewer for their feedback and questions! We’ve revised our paper to include additional discussion on the choice of latent dimension, visualizations of latent trajectories over coherent and incoherent text, and formatting/error edits raised by the reviewer.
>
> > The information about each of the ablations (ID, BM) could be explained better.
>
> Thanks for the suggestion! We’ve revised the paper to reflect the reviewer’s recommendations: In Appendix E, we elaborate in more depth with how the ablations are implemented.
>
> > Inconsistency in the best TC method between different latent dimensions [...] I wonder if you have thoughts about this.
>
> Thanks for raising this point! On tasks where the temporal structure isn’t clear (eg. ‘discourse structure’ on Wikisection, Table 1), the latent has a risk of picking spurious pieces of information, and this leads to uninformative latents. We think it’s an interesting open problem on how we might make the choice of latent dimension more robust!
>
> > Table 5 the VAE(32) method performs the best overall in "Wiki section" although the TC (16) method has been highlighted as the best. Is there a reason behind this?
>
> Thanks for catching this! That was an error on our side. We’ve updated the table to correct this mistake.
>
> > During the training of the decoder how do you make sure that the decoder uses the information given by the latent plan?
>
> Although there is no explicit guarantee that the decoder uses the information from the latent plan during fine tuning, we do see promising empirical results which suggest the latent plan is useful in training. For example, we see in Table 9 that Time Control uses latent information in Table 9 where PPL is lower with latent plans than just with fine tuning GPT-2 without latent plans.
>
> > Overall the paper would have benefited from an intrinsic visualization of the latent space [...] by visualizing the planning trajectory difference between coherent and incoherent text.
>
> Thanks for this suggestion! We’ve added a few latent trajectory examples over coherent and incoherent texts in Appendix J. For incoherent texts, we randomly mixed the sentence ordering of the original document. As the visualizations suggest, there is no information collapse of the embeddings. For completeness, we’ve also included trajectories of the ablated methods.

---

### Decision · Program_Chairs · 2022-01-20

**Decision:**

Accept (Oral)

**Comment:**

All reviewers found that the proposed LM with Brownian motion is interesting and novel. Several reviewers raised (minor) concerns about experiments, but have been generally resolved by the authors.